# YOUR CONTRASTIVE LEARNING IS SECRETLY DOING STOCHASTIC NEIGHBOR EMBEDDING

**Tianyang Hu[1], Zhili Liu[1,2], Fengwei Zhou[1], Wenjia Wang[2,3], Weiran Huang[1,4]\***
[1] Huawei Noah's Ark Lab, [2] Hong Kong University of Science and Technology
[3] Hong Kong University of Science and Technology (Guangzhou)
[4] Qing Yuan Research Institute, Shanghai Jiao Tong University

## ABSTRACT

Contrastive learning, especially self-supervised contrastive learning (SSCL), has achieved great success in extracting powerful features from unlabeled data. In this work, we contribute to the theoretical understanding of SSCL and uncover its connection to the classic data visualization method, stochastic neighbor embedding (SNE) (Hinton & Roweis, 2002), whose goal is to preserve pairwise distances. From the perspective of preserving neighboring information, SSCL can be viewed as a special case of SNE with the input space pairwise similarities specified by data augmentation. The established correspondence facilitates deeper theoretical understanding of learned features of SSCL, as well as methodological guidelines for practical improvement. Specifically, through the lens of SNE, we provide novel analysis on domain-agnostic augmentations, implicit bias and robustness of learned features. To illustrate the practical advantage, we demonstrate that the modifications from SNE to $t$-SNE (Van der Maaten & Hinton, 2008) can also be adopted in the SSCL setting, achieving significant improvement in both in-distribution and out-of-distribution generalization.

## 1 INTRODUCTION

Recently, contrastive learning, especially self-supervised contrastive learning (SSCL) has drawn massive attention, with many state-of-the-art models following this paradigm in both computer vision (He et al., 2020a; Chen et al., 2020a;b; Grill et al., 2020; Chen & He, 2021; Zbontar et al., 2021) and natural language processing (Fang et al., 2020; Wu et al., 2020; Giorgi et al., 2020; Gao et al., 2021; Yan et al., 2021). In contrast to supervised learning, SSCL learns the representation through a large number of unlabeled data and artificially defined self-supervision signals, i.e., regarding the augmented views of a data sample as positive pairs and randomly sampled data as negative pairs. By enforcing the features of positive pairs to align and those of negative pairs to be distant, SSCL produces discriminative features with state-of-the-art performance for various downstream tasks.

Despite the empirical success, the theoretical understanding is under-explored as to how the learned features depend on the data and augmentation, how different components in SSCL work and what are the implicit biases when there exist multiple empirical loss minimizers. For instance, SSCL methods are widely adopted for pretraining, whose feature mappings are to be utilized for various downstream tasks which are usually out-of-distribution (OOD). The distribution shift poses great challenges for the feature learning process with extra requirement for robustness and OOD generalization (Arjovsky et al., 2019; Krueger et al., 2021; Bai et al., 2021; He et al., 2020b; Zhao et al., 2023; Dong et al., 2022), which demands deeper understanding of the SSCL methods.

The goal of SSCL is to learn the feature representations from data. For this problem, one classic method is SNE (Hinton et al., 2006) and its various extensions. Specially, $t$-SNE (Van der Maaten & Hinton, 2008) has become the go-to choice for low-dimensional data visualization. Comparing to SSCL, SNE is far better explored in terms of theoretical understanding (Arora et al., 2018; Linderman & Steinerberger, 2019; Cai & Ma, 2021). However, its empirical performance is not satisfactory, especially in modern era where data are overly complicated. Both trying to learn feature representations, are there any deep connections between SSCL and SNE? Can SSCL take the advantage of the theoretical soundness of SNE? Can SNE be revived in the modern era by incorporating SSCL?

---

*Correspondence to Weiran Huang (weiran.huang@sjtu.edu.cn).

In this work, we give affirmative answers to the above questions and demonstrate how the connections to SNE can benefit the theoretical understandings of SSCL, as well as provide methodological guidelines for practical improvement. The main contributions are summarized below.

• We propose a novel perspective that interprets SSCL methods as a type of SNE methods with the aim of preserving pairwise similarities specified by the data augmentation.

• The discovered connection enables deeper understanding of SSCL methods. We provide novel theoretical insights for domain-agnostic data augmentation, implicit bias and OOD generalization. Specifically, we show isotropic random noise augmentation induces $l_2$ similarity while mixup noise can potentially adapt to low-dimensional structures of data; we investigate the implicit bias from the angle of order preserving and identified the connection between minimizing the expected Lipschitz constant of the SSCL feature map and SNE with uniformity constraint; we identify that the popular cosine similarity can be harmful for OOD generalization.

• Motivated by the SNE perspective, we propose several modifications to existing SSCL methods and demonstrate practical improvements. Besides a re-weighting scheme, we advocate to lose the spherical constraint for improved OOD performance and a $t$-SNE style matching for improved separation. Through comprehensive numerical experiments, we show that the modified $t$-SimCLR outperforms the baseline with 90% less feature dimensions on CIFAR-10 and $t$-MoCo-v2 pretrained on ImageNet significantly outperforms in various domain transfer and OOD tasks.

## 2 PRELIMINARY AND RELATED WORK

**Notations.** For a function $f : \Omega \to \mathbb{R}$, let $\|f\|_\infty = \sup_{\boldsymbol{x} \in \Omega} |f(\boldsymbol{x})|$ and $\|f\|_p = (\int_\Omega |f(\boldsymbol{x})|^p d\boldsymbol{x})^{1/p}$. For a vector $\boldsymbol{x}$, $\|\boldsymbol{x}\|_p$ denotes its $p$-norm, for $1 \le p \le \infty$. $\mathbb{P}(A)$ is the probability of event $A$. For a random variable $z$, we use $P_z$ and $p_z$ to denote its probability distribution and density respectively. Denote Gaussian distribution by $N(\mu, \Sigma)$ and let $\boldsymbol{I}_d$ be the $d \times d$ identity matrix. Let the dataset be $\mathcal{D}_n = \{\boldsymbol{x}_1, \cdots, \boldsymbol{x}_n\} \subset \mathbb{R}^d$ where each $\boldsymbol{x}_i$ independently follows distribution $P_{\boldsymbol{x}}$. The goal of unsupervised representation learning is to find informative low-dimensional features $\boldsymbol{z}_1, \cdots, \boldsymbol{z}_n \in \mathbb{R}^{d_z}$ of $\mathcal{D}_n$ where $d_z$ is usually much smaller than $d$. We use $f(\boldsymbol{x})$ to as the default notation for the feature mapping from $\mathbb{R}^d \to \mathbb{R}^{d_z}$, i.e., $\boldsymbol{z}_i = f(\boldsymbol{x}_i)$.

**Stochastic neighbor embedding.** SNE (Hinton & Roweis, 2002) is a powerful representation learning framework designed for visualizing high-dimensional data in low dimensions by preserving neighboring information. The training process can be conceptually decomposed into the following two steps: (1) calculate the pairwise similarity matrix $\boldsymbol{P} \in \mathbb{R}^{n \times n}$ for $\mathcal{D}_n$; (2) optimize features $\boldsymbol{z}_1, \cdots, \boldsymbol{z}_n$ such that their pairwise similarity matrix $\boldsymbol{Q} \in \mathbb{R}^{n \times n}$ matches $\boldsymbol{P}$. Under the general guidelines lie plentiful details. In Hinton & Roweis (2002), the pairwise similarity is modeled as conditional probabilities of $\boldsymbol{x}_j$ being the neighbor of $\boldsymbol{x}_i$, which is specified by a Gaussian distribution centered at $\boldsymbol{x}_i$, i.e., when $i \ne j$,

$$P_{j|i} = \frac{\exp(-\|\boldsymbol{x}_i - \boldsymbol{x}_j\|_2^2 / 2\sigma_i^2)}{\sum_{k \ne i} \exp(-\|\boldsymbol{x}_i - \boldsymbol{x}_k\|_2^2 / 2\sigma_i^2)}, \tag{2.1}$$

where $\sigma_i$ is the variance of the Gaussian centered at $\boldsymbol{x}_i$. Similar conditional probabilities $Q_{j|i}$'s can be defined on the feature space. When matching $\boldsymbol{Q}$ to $\boldsymbol{P}$, the measurement chosen is the KL-divergence between two conditional probabilities. The overall training objective for SNE is

$$\inf_{\boldsymbol{z}_1, \cdots, \boldsymbol{z}_n} \sum_{i=1}^n \sum_{j=1}^n P_{j|i} \log \frac{P_{j|i}}{Q_{j|i}}. \tag{2.2}$$

Significant improvements have been made to the classic SNE. Im et al. (2018) generalized the KL-divergence to $f$-divergence and found that different divergences favors different types of structure. Lu et al. (2019) proposed to make $P$ doubly stochastic so that features are less crowded. Most notably, $t$-SNE (Van der Maaten & Hinton, 2008) modified the pairwise similarity by considering joint distribution rather than conditional, and utilizes t-distribution instead of Gaussian in the feature space modeling. It is worth noting that SNE belongs to a large class of methods called manifold learning (Li et al., 2022). In this work, we specifically consider SNE. If no confusion arises, we use SNE to denote the specific work of Hinton & Roweis (2002) and this type of methods in general interchangeably.

**Self-supervised contrastive learning.** The key part of SSCL is the construction of positive pairs, or usually referred to as different views of the same sample. For each $\boldsymbol{x}_i$ in the training data, denote

its two augmented views to be $\boldsymbol{x}_i'$ and $\boldsymbol{x}_i''$. Let $\mathcal{D}_n' = \{\boldsymbol{x}_1', \cdots, \boldsymbol{x}_n'\}$, $\mathcal{D}_n'' = \{\boldsymbol{x}_1'', \cdots, \boldsymbol{x}_n''\}$ and define

$$l(\boldsymbol{x}_i', \boldsymbol{x}_i'') = -\log \frac{\exp(\mathrm{sim}(f(\boldsymbol{x}_i'), f(\boldsymbol{x}_i''))/\tau)}{\sum_{\boldsymbol{x} \in \mathcal{D}_n' \cup \mathcal{D}_n'' \setminus \{\boldsymbol{x}_i'\}} \exp(\mathrm{sim}(f(\boldsymbol{x}_i'), f(\boldsymbol{x}))/\tau)},$$

where $\mathrm{sim}(\boldsymbol{z}_1, \boldsymbol{z}_2) = \langle \frac{\boldsymbol{z}_1}{\|\boldsymbol{z}_1\|_2}, \frac{\boldsymbol{z}_2}{\|\boldsymbol{z}_2\|_2} \rangle$ denotes the cosine similarity and $\tau$ is a temperature parameter. The training objective of the popular SimCLR (Chen et al., 2020a) can be written as $L_{\mathrm{InfoNCE}} := \frac{1}{2n} \sum_{i=1}^{n} (l(\boldsymbol{x}_i'', \boldsymbol{x}_i') + l(\boldsymbol{x}_i', \boldsymbol{x}_i''))$.

Recently, various algorithms are proposed to improve the above contrastive learning. To address the need for the large batch size, MoCo (He et al., 2020a; Chen et al., 2020b) utilizes a moving-averaged encoder and a dynamic memory bank to store negative representations, making it more device-friendly. Grill et al. (2020); Chen & He (2021); Zbontar et al. (2021); Chen et al. (2021) radically discard negative samples in SSCL but still achieve satisfactory transfer performance. Another line of works (Caron et al., 2020; Li et al., 2021; Liu et al., 2022) mines the hierarchy information in data to derive more semantically compact representations. Radford et al. (2021); Yao et al. (2021) even extend the contrastive methods to the multi-modality data structure to achieve impressive zero-shot classification results.

**Theoretical understanding of SSCL.** In contrast of the empirical success, theoretical understanding of SSCL is still limited. While most of theoretical works (Arora et al., 2019; Tosh et al., 2020; HaoChen et al., 2021; 2022; Wang et al., 2022; Wen & Li, 2021; Wei et al., 2020; Huang et al., 2021; Ji et al., 2021; Ma et al., 2023) focus on its generalization ability on downstream tasks, there are some works studying specifically the InfoNCE loss. One line of works (Oord et al., 2018; Bachman et al., 2019; Hjelm et al., 2018; Tian et al., 2019; 2020) understand the InfoNCE loss from mutual information perspective, showing that the negative InfoNCE is a lower bound of mutual information between positive samples. Other works (Wang & Isola, 2020; Huang et al., 2021; Jing et al., 2021) are from the perspective of geometry of embedding space, showing that InfoNCE can be divided into two parts: one controls alignment and the other prevents representation collapse. In this paper, we study SSCL from the SNE perspective, which, to the best of the authors' knowledge, has no discussion in existing literature. The closest work to ours is Balestriero & LeCun (2022), which proposed a unifying framework under the helm of spectral manifold learning. In comparison, our work focus specifically on the connection between SSCL and SNE.

## 3  SNE PERSPECTIVE OF SSCL

A closer look at the training objectives of SNE and SimCLR reveals great resemblance — SimCLR can be seen as a special SNE model. To see this, denote $\widetilde{\mathcal{D}}_{2n} = \mathcal{D}_n'' \cup \mathcal{D}_n'$ as the augmented dataset with index $\widetilde{\boldsymbol{x}}_{2i-1} = \boldsymbol{x}_i''$ and $\widetilde{\boldsymbol{x}}_{2i} = \boldsymbol{x}_i'$. If we change the $l_2$ distance to the negative cosine similarity and let $\sigma_i^2 \equiv \tau$. Admitting similar conditional probability formulation as in (2.1) yields that for $i \neq j$,

$$\widetilde{Q}_{j|i} = \frac{\exp(\mathrm{sim}(f(\widetilde{\boldsymbol{x}}_i), f(\widetilde{\boldsymbol{x}}_j))/\tau)}{\sum_{k \neq i} \exp(\mathrm{sim}(f(\widetilde{\boldsymbol{x}}_i), f(\widetilde{\boldsymbol{x}}_k))/\tau)}. \tag{3.1}$$

By taking

$$\widetilde{P}_{j|i} = \begin{cases} 1, & \text{if } \widetilde{\boldsymbol{x}}_i \text{ and } \widetilde{\boldsymbol{x}}_j \text{ are positive pairs} \\ 0, & \text{otherwise,} \end{cases} \tag{3.2}$$

the SNE objective (2.2) can be written as

$$\sum_{i=1}^{2n} \sum_{j=1}^{2n} \widetilde{P}_{j|i} \log \frac{\widetilde{P}_{j|i}}{\widetilde{Q}_{j|i}} = \sum_{k=1}^{n} \left( -\log(\widetilde{Q}_{2k-1|2k}) - \log(\widetilde{Q}_{2k|2k-1}) \right),$$

which reduces to the SimCLR objective $L_{\mathrm{InfoNCE}}$, up to a constant scaling term only depending on $n$.

Now that we have established the correspondence between SNE and SimCLR, it's clear that the feature learning process of SSCL also follows the two steps of SNE.

(S1) The positive pair construction specifies the similarity matrix $\boldsymbol{P}$.

(S2) The training process then matches $\boldsymbol{Q}$ to $\boldsymbol{P}$ by minimizing some divergence between the two specified by the training objective, e.g., KL divergence in SimCLR.

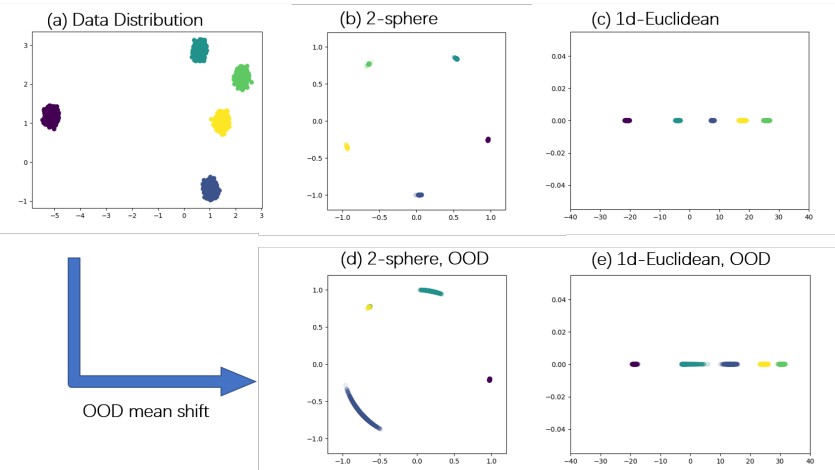

Figure 1: Gaussian mixture setting with 5 components. (a) illustration of data with 250 samples. (b) learned features by standard SimCLR with normalization (cosine similarity) to 1-sphere. (c) learned features by modified SimCLR without normalization ($l_2$ similarity). (d, e) feature mapping of the two methods in case of OOD mean shift. The linear classification accuracy is 48.4% in (d) and 100% in (e).

The main difference between SNE and SSCL is the first part, where the $P$ in SNE is usually densely filled by $l_p$ distance, ignoring the semantic information within rich data like images and texts. In contrast, SSCL omits all traditional distances in $\mathbb{R}^d$ and only specifies semantic similarity through data augmentations, and the resulting $P$ is sparsely filled only by positive pairs as in (3.2). For structurally rich data such as image or text, the semantic information is invariant to a wide range of transformations. Human's prior knowledge of such invariance guides the construction of positive pairs in SSCL, which is then learned by the feature mapping.

**Remark 3.1** (SNE vs SSCL). We would like to clarify on the main difference between SNE and SSCL that we focus in this work. Although standard SNE (Hinton et al., 2006) is *non-parametric* without explicit feature maps, and is optimized for the *whole dataset*, these are not the defining properties of SNE. SNE can also utilize explicit feature maps and mini-batch training (Van Der Maaten, 2009). On the other hand, SSCL can also benefit from larger/full batches (Chen et al., 2020a) and can also be modified to directly optimize the features $z_i$'s. In this work, we omit these subtleties[1] and focus on the (S1) perspective, which we view as the most significant difference between SNE and SSCL.

### 3.1 ANALYSIS

In this section, to showcase the utility of the SNE perspective, we demonstrate how the feature learning process of SSCL methods, e.g., SimCLR, can become more intuitive and transparent. Specifically, we re-derive the alignment and uniformity principle (Wang & Isola, 2020) as well as provide novel analysis on domain-agnostic augmentations, the implicit bias and robustness of learned features. To aid the illustration, we device toy examples with simulated Gaussian mixture data.

**Gaussian mixture setting.** Let the data follow $d$-dimensional Gaussian mixture distribution with $m$ components where $P_{\boldsymbol{x}} \sim \frac{1}{m} \sum_{i=1}^{m} N(\boldsymbol{\mu}_i, \sigma^2 \boldsymbol{I}_d)$. The special case with $d = 2$, $m = 5$, $\sigma = 0.1$ is illustrated in Figure 1(a) with 250 independent samples. To apply contrastive methods, consider constructing positive pairs by direct sampling, i.e., if $\boldsymbol{x}$ is from the first component, then we sample another $\boldsymbol{x}' \sim N(\boldsymbol{\mu}_1, \sigma^2 \boldsymbol{I}_d)$ independently as its alternative view for contrast. The negative samples are the same as in standard SimCLR training.

#### 3.1.1 DOMAIN-AGNOSTIC DATA AUGMENTATION

Now that we have established in (S1) that the input space pairwise distance is specified by the data augmentation, a natural question to ask is what are the corresponding *induced distances*. In this section, we investigate this problem for domain-agnostic data augmentations.

The quality of data augmentation has great impact on the performance of SSCL methods, which reflects people's prior knowledge on the data. However, when facing new data without any domain knowledge,

---

[1]All the contrastive losses are written in full batches for simplicity in this work as we focus on analyzing the optimal solutions of SSCL methods rather than the optimization process.

we have to rely on domain-agnostic data augmentations, e.g., adding random noises (Verma et al., 2021), for contrast. We first consider using general random noise augmentation, i.e., for any $\boldsymbol{x} \in \mathbb{R}^d$, let $\boldsymbol{x}' = \boldsymbol{x} + \delta$ where $\delta$ follows some distribution with density $\phi(\boldsymbol{x})$. Then, for any $\boldsymbol{x}_i$, the probability density of having $\boldsymbol{t} \in \mathbb{R}^d$ as its augmented point can be characterized as $P_{\boldsymbol{t}|\boldsymbol{x}_i} = \mathbb{P}(\boldsymbol{x}_i \text{ and } \boldsymbol{x}'_i = \boldsymbol{t} \text{ form a positive pair}|\boldsymbol{x}_i) = \phi(\boldsymbol{t} - \boldsymbol{x}_i)$. We have the following proposition on Gaussian-induced distance.

**Proposition 3.2** (Gaussian noise injection). *If the noise distribution is isotropic Gaussian with mean zero, the induced distance is* equivalent *to the $l_2$ distance in $\mathbb{R}^d$, up to a monotone transformation.*

Another popular noise injection method is the mixup (Zhang et al., 2017), where the augmented data are comprised of convex combinations of the training data. For each $\boldsymbol{x}_i$, a positive pair can be constructed from another $\boldsymbol{x}_j$ such that $\boldsymbol{x}'_i = \boldsymbol{x}_i + \lambda(\boldsymbol{x}_j - \boldsymbol{x}_i)$ and $\lambda \in (0,1)$ is the hyperparameter usually modeled with Beta distribution. For independent $\boldsymbol{x}_1, \boldsymbol{x}_2 \sim P_x$, denote the convoluted density of $\lambda(\boldsymbol{x}_1 - \boldsymbol{x}_2)$ as $p_\lambda(\boldsymbol{x})$, which is symmetric around 0. Then, if employing mixup for positive pairs in SSCL, the induced distance can be written as $P_{\boldsymbol{x}_1, \boldsymbol{x}_2} = P_{\boldsymbol{x}_2, \boldsymbol{x}_1} = p_\lambda(\boldsymbol{x}_1 - \boldsymbol{x}_2)$.

**Gaussian vs. mixup.** Verma et al. (2021) proposed to use mixup when domain-specific information is unattainable and provided supportive analysis on its advantage over isotropic Gaussian noise from the classification generalization error point of view. Through (S1) perspective, we can intuitively explain why data-dependent mixup noises can be potentially better from the perspective of the "*curse of dimensionality*". Consider the $d$-dimensional Gaussian mixture setting with $m < d$ separated components. Notice that $\boldsymbol{\mu}_1, \cdots, \boldsymbol{\mu}_m$ can take up at most $(m-1)$-dimensional linear sub-space of $\mathbb{R}^d$. Denoted the space spanned by $\boldsymbol{\mu}_i$'s as $\boldsymbol{S}_\mu$. For the light-tailed Gaussian distribution, and the majority of samples will be close to $\boldsymbol{S}_\mu$. Hence, majority of the convoluted density $p_\lambda(\boldsymbol{x})$ will also be supported on $\boldsymbol{S}_\mu$, so does the corresponding $P_{\boldsymbol{x}_2, \boldsymbol{x}_1}$. Thus, the induced distance from mixup will omit irrelevant variations in the complement of $\boldsymbol{S}_\mu$ and focus on the low-dimensional sub-space $\boldsymbol{S}_\mu$ where $\boldsymbol{\mu}_i$'s actually differ. This effectively reduces the dimension dependence from $d$ to $m-1$. In comparison, isotropic Gaussian noise induces $l_2$ distance for positive pairs with support of $\mathbb{R}^d$, which will be much more inefficient, especially when $m \ll d$. Since it is well-known that the performance of regression or classification models is strongly influenced by the intrinsic dimension of the input space (Hamm & Steinwart, 2021), keeping the data in a low-dimensional space is preferable.

### 3.1.2 ALIGNMENT AND UNIFORMITY

Characterizing the learned features of SSCL is of critical importance. Wang & Isola (2020) proposed alignment and uniformity as principles for SimCLR type contrastive learning methods. Such results can be intuitively understood through the perspective of (S1) and (S2). Consider the common case where the feature space is $(d_z - 1)$-sphere. First, (3.2) indicates that only similarities (distances) between positive pairs are non-zero (finite) and all other pairwise similarities (distances) are zero (infinity). Preserving (3.2) requires the features of positive pairs to align (cosine similarity tends to 1) and those of negative pairs to be as distant as possible. If in the extreme case where positive pairs match exactly, i.e., $f(\boldsymbol{x}_i) = f(\boldsymbol{x}'_i)$ for any $i = 1, \cdots, n$, we call it *perfect alignment*.

If perfect alignment is achieved and the features are constrained on the unit sphere, matching (3.2) implies pushing $n$ points on the feature space as distant as possible. Maximally separated $n$ points on a $d$-sphere has been studied in geometry, known as the Tammes problem (Tammes, 1930; Erber & Hockney, 1991; Melisseny, 1998). We say *perfect uniformity* is achieved if all the pairs are maximally separated on the sphere. There are some simple cases of the Tammes problem. If $d = 2$, perfect uniformity can be achieved if the mapped points form a regular polygon. If $d \geq n-1$, the solution can be given by the vertices of an $(n-1)$-simplex, inscribed in an $(n-1)$-sphere embedded in $\mathbb{R}^d$. The cosine similarity between any two vertices is $-1/(n-1)$ and in this case, $L_{\text{InfoNCE}}$ can attain its lower bound[2]. As $n \to \infty$, the point distribution converges weakly to uniform distribution. As can be seen in Figure 1(a, b), perfect alignment and perfect uniformity are almost achieved by standard SimCLR in the Gaussian mixture setting.

As we will demonstrate in Section 3.1.4 that the spherical feature space can be bad for OOD generalization, adopting of the Euclidean space will change the statement of the uniformity property and can also be analyzed from the SNE perspective. Details can be found in Appendix A.5.

---

[2]Notice that in this case, the optimal feature mapping will contain little information of the data, mapping anchor samples to interchangeable points with identical pairwise distances

### 3.1.3 IMPLICIT BIAS

Existing theoretical results on SSCL provide justification of its empirical success in classification. However, there is more to it than just separating different classes and many phenomena are left unexplained. Take the popular SimCLR (Chen et al., 2020a) on CIFAR-10 as an example, we can consistently observe that the feature similarities within animals (bird, cat, deer, dog, frog, horse) and within objects (airplane, automobile, ship, truck), are significantly higher than those between animals and objects[3]. This can be viewed as an implicit bias towards preserving semantic information, which might be surprising as we have no supervision on the label information during the training process. However, existing literature on implicit bias is scarce. As advocated in Saunshi et al. (2022), ignoring inductive biases cannot adequately explain the success of contrastive learning. In this section, we provide a simple explanation from the perspective of SNE.

For a more concrete illustration, consider training SimCLR in the Gaussian mixture setting with $d = 1, d_z = 2, m = 4, \mu_i = i$, and $\sigma = 0.1$. Denote the 4 components in ascending order by A,B,C,D. Perfect alignment and uniformity imply that their feature maps (a, b, c, d) on the unit-circle should be vertices of an inscribed square. What left unsaid is their *relative order*. Clockwise or counter-Clockwise from a, regardless of the initialization, we can observe SimCLR to consistently produce the order a $\to$ b $\to$ c $\to$ d.

**Remark 3.3** (Relative ordering and neighbor-preserving)**.** The order-preserving property showcased with $d = 1$ is mainly for illustration, as in one-dimension, the neighboring info is simplified as the order, which is much easier to understand. The results remain the same in high dimensions as long as the clusters are well separated with an obvious order of clusters. For instance, some relative orders in Figure 1(a,b) are also stable, e.g., the neighbor of blue will consistently be purple and yellow.

With great resemblance to SNE, SSCL methods also exhibit neighbor-preserving property and we identify it as an implicit bias. Such implicit bias can be universal in SSCL and the phenomenon in Figure A.3 is also a manifestation. In deep learning, the implicit bias is usually characterized by either closeness to the initialization (Moroshko et al., 2020; Azulay et al., 2021), or minimizing certain complexity (Razin & Cohen, 2020; Zhang et al., 2021). In the case of SimCLR, we hypothesize the implicit bias as the *expected Lipschitz constant*, which has deep connections to SNE with uniformity constraint. For a feature map $f$ onto the unit-sphere, define

$$C(f) = \mathbb{E}_{\boldsymbol{x}, \boldsymbol{x}'} \frac{\|f(\boldsymbol{x}) - f(\boldsymbol{x}')\|_2}{\|\boldsymbol{x} - \boldsymbol{x}'\|_2}, \tag{3.3}$$

where the $\boldsymbol{x}_1, \boldsymbol{x}_2$ are independent samples from the data distribution.

**Definition 3.4** (SNE with uniformity constraint)**.** Assume data $\boldsymbol{x}_1, \cdots, \boldsymbol{x}_n \in \mathbb{R}^d$. If the corresponding SNE features $\boldsymbol{z}_1, \cdots, \boldsymbol{z}_n \in \mathbb{R}^{d_z}$ are constrained to be the maximally separated $n$ points on the $(d_z - 1)$-sphere, we call this problem *SNE with uniformity constraint*.

The key of SNE is matching the pairwise similarity matrices $Q$ to $P$. When solving SNE with uniformity constraint, the only thing to be optimized is the pairwise correspondence, or ordering of the mapping. We have the following theorem that links the neighbor-preserving property to $C(f)$.

**Theorem 3.5.** Let $\boldsymbol{x}_1, \cdots, \boldsymbol{x}_n \in \mathbb{R}^d$ such that $\|\boldsymbol{x}_i - \boldsymbol{x}_j\|_2 > 0$ for any $i, j$ and let $\boldsymbol{z}_1, \cdots, \boldsymbol{z}_n \in \mathbb{R}^{d_z}$ be maximally separated $n$ points on the $(d_z - 1)$-sphere. Denote $P = (p_{ij})_{n \times n}$ and $Q = (q_{ij})_{n \times n}$ as the corresponding pairwise similarity matrices of $\boldsymbol{x}_i$'s and $\boldsymbol{z}_i$'s respectively. Let $\pi$ denote a permutation on $\{1, \cdots, n\}$ and denote all such permutations as $T$. Let $Q^\pi$ as the $\pi$-permuted matrix $Q$ and define

$$C_1(P, Q^\pi) = \sum_{i \neq j} \frac{q_{\pi(i)\pi(j)}}{p_{ij}} \quad \text{and} \quad \pi^* = \underset{\pi \in T}{\operatorname{argmin}} \, C_1(P, Q^\pi).$$

Then, $\pi^*$ also minimizes $\|\bar{P} - Q^\pi\|_F$ where $\|\cdot\|_F$ is the Frobenius norm and $\bar{P} = (\bar{p}_{ij})_{n \times n}$ is a (monotonically) transformed similarity matrix with $\bar{p}_{ij} = -1/p_{ij}$.

Theorem 3.5 showcases the relationship between minimizing $C(f)$ and the structure preserving property by considering a special SNE problem, where the pairwise similarity is not modeled by Gaussian as standard. Although $q_{ij} = -\|f(\boldsymbol{x}_i) - f(\boldsymbol{x}_j)\|_2$ is unorthodox, it is reasonable since the larger the distance, the smaller the similarity. We have the following corollary to explain the neighbor-preserving property of SSCL and the implicit bias associated with minimizing the complexity $C(f)$.

---

[3]Figure A.3 illustrates the phenomenon. Details can be found in Appendix A.1

**Corollary 3.6** (Implicit bias of SSCL). When SSCL model achieves perfect alignment and perfect uniformity, if the complexity $C(f)$ is minimized, the resulting feature map preserves pairwise distance in the input space, resembling SNE with uniformity constraint.

Corollary 3.6 links the implicit bias of SSCL to the SNE optimization with uniformity constraint. In the case of perfect alignment and perfect uniformity, SSCL can be seen as a special SNE problem where the feature $z_1, \cdots, z_n$ must be maximally separated on the unit-sphere. Recall the 1-dimension Gaussian case. There are in total $3! = 6$ different orderings for the 4 cluster means, among which, a $\rightarrow b \rightarrow c \rightarrow d$ will give the lowest SNE loss. As can be seen in Figure A.4, both $C(f)$ and the SNE loss are monotonically decreasing during training for the Gaussian mixture setting.

When the alignment or uniformity is not perfect, the resulting feature mapping can still be characterized via SNE, with the uniformity constraint relaxed as a form of regularization. In our numerical experiments on the CIFAR-10 data, we observe $C(f)$ to be monotonically decreasing during the training process, supporting our hypothesis. More details can be found in Appendix A.3. Corollary 3.6 sheds light on the implicit semantic information preserving phenomenon shown in Figure A.3, as in the input space, images of dogs should be closer to images of cats, than airplanes.

### 3.1.4 TARGETING OOD: EUCLIDEAN VS SPHERICAL

Almost all SSCL methods require normalization to the unit-sphere and the similarity on the feature space is often the cosine similarity. In comparison, standard SNE methods operate freely on the Euclidean space. In this section, we show that the normalization can hinder the structure-preserving and there is a fundamental *trade off* between in-distribution and out-of-domain generalization.

Consider the 2-dimensional Gaussian mixture setting as illustrated in Figure 1(a). Notice that as long as the mixing components are well separated, the learned feature mapping on the sphere will always be the pentagon shape, regardless of the relative locations of the clusters. This is a result of the uniformity property derived under spherical constraint. Distant clusters in the input space will be pulled closer while close clusters will be pushed to be more distant, which results in the trade off between in-distribution and out-of-domain generalization. On one hand, close clusters being more separated in the feature space is potentially beneficial for in-distribution classification. On the other hand, the spherical constraint adds to the complexity of the feature mapping, potentially hurting robustness.

In the Euclidean space, pushing away negative samples (as distant as possible) will be much easier, since the feature vectors could diverge towards infinity[4] and potentially preserve more structural information. To verify our intuition, we relax the spherical constraint in the Gaussian mixture setting and change the cosine similarity in SimCLR to the negative $l_2$ distance in $\mathbb{R}$. The learned features are shown in Figure 1(c). Comparing to Figure 1(b), we can get the extra information that the purple cluster is far away to the others. If we introduce a small mean shift to the data, moving the distribution along each dimension by 1, the resulting feature maps differ significantly in robustness. As illustrated in Figure 1(d) vs. (e), the standard SimCLR are much less robust to OOD shifts and the resulting classification accuracy degrades to only 48.4%, while that for the modified SimCLR remains 100%. The same OOD advantage can also be verified in the CIFAR-10 to CIFAR-100 OOD generalization case (details in Appendix C.3 Figure C.8) and large-scale real-world scenarios with MoCo (Chen et al., 2020b) as baseline (details in Section 5).

## 4 IMPROVING SSCL BY SNE

The proposed SNE perspective (S1,S2) can inspire various modifications to existing SSCL methods. In this section, we choose SimCLR as our baseline and investigate three straightforward modifications. For empirical evaluation, we report the test classification accuracy of nearest neighbor classifiers on both simulated data and real datasets. Experiment details can be found in Appendix C.

### 4.1 WEIGHTED POSITIVE PAIRS

In practice, positive pairs are constructed from anchors (training data), by i.i.d. data augmentations, e.g., random resized crop, random horizontal flip, color jitter, etc. Take random crop as an example, pair 1 and 2 may be from 30%, 80% random crops, respectively. Their similarities should not be treated

---

[4]In practice, various regularization, e.g, weight decay, are employed and the resulting features will be bounded.

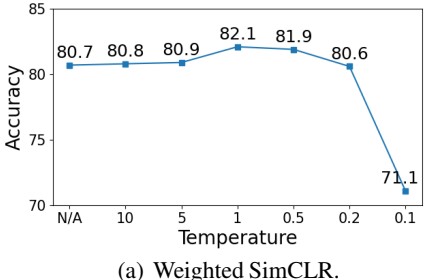
(a) Weighted SimCLR.

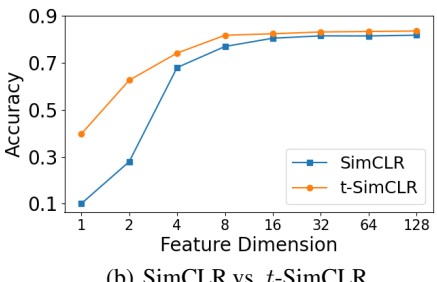
(b) SimCLR vs. $t$-SimCLR.

Figure 2: Nearest neighbor classification test accuracy on CIFAR-10 with ResNet-18 after 200 epochs pre-training. (a) "N/A" stands for the baseline SimCLR. The $x$-axis is the temperature for IoU weighting scheme. (b) Comparison between SimCLR and $t$-SimCLR with different feature dimensions.

as equal, as in typical SSCL methods. Incorporating the disparity in the data augmentation process is straightforward in the perspective of SNE, where the InfoNCE loss can be naturally modified as

$$\frac{1}{2n}\sum_{i=1}^{n} p_{ii'} \cdot (l(\boldsymbol{x}_i, \boldsymbol{x}_i') + l(\boldsymbol{x}_i', \boldsymbol{x}_i)).$$

The weight $p_{ii'}$ in $P$ can be specified manually to reflect human's prior knowledge. To test out the effect of such modification, we conduct numerical experiments on CIFAR-10 using the standard SimCLR. The weighting scheme is based on the Intersection over Union (IoU) of random resized crops. For each positive pair, let $p_{ii'} \propto \exp(\mathrm{IoU}(\boldsymbol{x}_i, \boldsymbol{x}_i')/\tau')$, where $\tau' > 0$ is a hyperparameter (temperature) controlling the strength of the weighting scheme, i.e., the bigger the $\tau'$, the closer to the unweighted state. The CIFAR-10 test performance vs. $\tau'$ is shown in Figure 2(a). The baseline is 80.7% and can be significantly improved to 82.1% if choosing $\tau' = 1$.

## 4.2 T-SIMCLR: $t$-SNE STYLE MATCHING

Most SSCL algorithms differ mainly in (S2), i.e., defining $\boldsymbol{Q}$ and matching it to $\boldsymbol{P}$, where fruitful results in SNE literature can be mirrored and applied. Now that we have identified the advantage of modeling features in Euclidean spaces in Section 3.1.4, the most promising modification that follows is to introduce $t$-SNE to SimCLR. Since we are learning low-dimensional features from high-dimensional data, preserving all pairwise similarities is impossible and the features tend to collapse. This is referred to as the *"crowding problem"* in Van der Maaten & Hinton (2008) (see Section 3.2 therein). $t$-SNE utilizes the heavy-tail t-distribution instead of the light-tail Gaussian, to model $\boldsymbol{Q}$ and encourage separation in feature space. Correspondingly, the training objective $L_{\mathrm{InfoNCE}}$ can be modified as

$$\frac{1}{n}\sum_{i=1}^{n} -\log \frac{\left(1+\|f(\boldsymbol{x}_i)-f(\boldsymbol{x}_i')\|_2^2/(\tau\, t_{df})\right)^{-(t_{df}+1)/2}}{\sum_{1\le j\ne k\le 2n}(1+\|f(\widetilde{\boldsymbol{x}}_j)-f(\widetilde{\boldsymbol{x}}_k)\|_2^2/(\tau\, t_{df}))^{-(t_{df}+1)/2}}, \tag{4.1}$$

where $t_{df}$ is the degree of freedom for the $t$-distribution. Besides substituting the cosine similarity to the $l_2$ distance, the key modification is the modeling of feature space similarity $\boldsymbol{Q}$, from Gaussian to $t$-distribution as suggested by Van der Maaten & Hinton (2008) to avoid the crowding problem and accommodate the dimension-deficiency in the feature space. We call the modified method $t$-*SimCLR* and we expect it to work better, especially when the feature dimension is low, or in the OOD case.

Figure 2(b) shows the comparison between SimCLR and $t$-SimCLR on CIFAR-10 with different feature dimensions, where $t$-SimCLR has significant advantages in all cases and the smaller the $d_z$, the larger the gap. Without decreasing the standard $d_z = 128$, $t$-SimCLR improves the baseline from 80.8% to 83.9% and even beats it using only $d_z = 8$ with accuracy 81.7%.

**Remark 4.1** (Degree of freedom). Standard $t$-SNE utilizes $t$-distribution with $t_{df} = 1$, to better accommodate the extreme $d_z = 2$ case. In practice, $t_{df}$ can vary and as $d_z$ increases, larger $t_{df}$ might be preferred. We recommend using $t_{df} = 5$ as the default choice. The performance of $t_{df}$ vs $d_z$ can be found in Appendix C, as well as discussion on the fundamental difference between $t_{df}$ and $\tau$.

**Remark 4.2** (Training epochs). For the CIFAR-10 experiments, we reported the results of ResNet-18 after 200 training epochs, similar to the setting of Yeh et al. (2021). We also conducted 1000-epoch experiments and found that our modifications provide consistent improvements throughout the training process, not in terms of speeding up the convergence, but converging to better solutions. Details can be found in Appendix C.1 and Figure C.6.

Table 1: Domain transfer results of vanilla MoCo-v2 and $t$-MoCo-v2.

| Method | Aircraft | Birdsnap | Caltech101 | Cars | CIFAR10 | CIFAR100 | DTD | Pets | SUN397 | Avg. |
|---|---|---|---|---|---|---|---|---|---|---|
| MoCo-v2 | 82.75 | 44.53 | 83.31 | 85.24 | 95.81 | 72.75 | **71.22** | 86.70 | 56.05 | 75.37 |
| $t$-MoCo-v2 | **82.78** | **53.46** | **86.81** | **86.17** | **96.04** | **78.32** | 69.20 | **87.95** | **59.30** | **77.78** |

Table 2: OOD accuracies of vanilla MoCo-v2 and $t$-MoCo-v2 on domain generalization benchmarks.

| Method | PACS | VLCS | Office-Home | Avg. |
|---|---|---|---|---|
| MoCo-v2 | 58.5 | 70.4 | 36.6 | 55.2 |
| $t$-MoCo-v2 | **61.3** | **75.1** | **42.1** | **59.5** |

## 5 LARGE SCALE EXPERIMENTS

In this section, we apply the same modifications proposed in Section 4.2 to MoCo-v2 (Chen et al., 2020b), as it is more device-friendly to conduct large scale experiments. We name our model **$t$-MoCo-v2**. Both models are pre-trained for 200 epochs on ImageNet following the setting of Chen et al. (2020b). The linear probing accuracy of $t$-MoCo-v2 on ImageNet is 67.0%, which is comparable to the MoCo result 67.5%. With the same level of in-distribution classification accuracy, we conduct extensive experiments to compare their OOD performance. The results in Table 1 and 2 suggest that our modification significantly improves the domain transfer and the OOD generalization ability without sacrificing in-distribution accuracy.

**Domain Transfer.** We first conduct experiments on the traditional self-supervision domain transfer benchmark. We compare MoCo-v2 and $t$-MoCo-v2 on Aircraft, Birdsnap, Caltech101, Cars, CIFAR10, CIFAR100, DTD, Pets, and SUN397. We follow transfer settings in Ericsson et al. (2021) to finetune the pre-trained models. The results are reported in Table 1. Our model $t$-MoCo-v2 surpasses MoCo-v2 in 8 out of 9 datasets, showing a significantly stronger transfer ability. Notice that our model is pre-trained with 200 epochs, surprisingly, compared with the original MoCo-v2 model pre-trained with 800 epochs, the fine-tuning results of $t$-MoCo-v2 are still better on Birdsnap, Caltech101, CIFAR100, and SUN397.

**Out-of-domain generalization.** As illustrated in Section 3.1.4, standard SSCL methods, e.g., SimCLR, MoCo, etc., could suffer from OOD shift. To demonstrate the advantage of our modification, we investigate the effectiveness of our method on OOD generalization benchmarks: PACS Li et al. (2017), VLCS Fang et al. (2013), Office-Home Venkateswara et al. (2017). We follow the standard way to conduct the experiment, i.e., choosing one domain as the test domain and using the remaining domains as training domains, which is named the leave-one-domain-out protocol. As can be seen in Table 2, our $t$-MoCo-v2 indicates significant improvement over MoCo-v2. Both experiments indicate our modification exhibits substantial enhancement for domain transfer and OOD generalization ability. Similar to domain transfer scenario, compared with the original MoCo-v2 model pre-trained with 800 epochs, $t$-MoCo-v2 is better on all of the three datasets. More experiment details, including detailed comparisons, are in Appendix C.

## 6 DISCUSSION

This work proposes a novel perspective that interprets SSCL methods as a type of SNE methods, which facilitates both deeper theoretical understandings and methodological guidelines for practical improvement. More interpretations of SSCL from preserving the distance between distributions can be found in Appendix B. Our analysis has limitations and the insights from SNE are not universally applicable for all SSCL methods, e.g., Zbontar et al. (2021); Yang et al. (2021) don't fit in our framework. However, this work is an interesting addition to existing theoretical works of SSCL and more investigations can be made along this path. While there are various extensions of the classic SNE, in this work, as a proof of concept, we mainly showcased practical improvements from $t$-SNE. We expect more modifications can be developed by borrowing advances in the SNE literature, e.g., changing to $f$-divergences (Im et al., 2018) or consider optimal transport Bunne et al. (2019); Salmona et al. (2021); Mialon et al. (2020). On the other hand, standard SNE methods can also borrow existing techniques in SSCL to improve their performance on more complicated data, e.g., incorporating data augmentations instead of or on top of pre-defined distances. In this sense, by choosing feature dimension to be 2, various SSCL methods can also be used as data visualization tools (Böhm et al., 2022; Damrich et al., 2022). Specifically on CIFAR-10, standard $t$-SNE can barely reveal any clusters while our $t$-SimCLR with $d_z = 2$ produces much more separation among different labels. More details can be found in Appendix C.7.

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

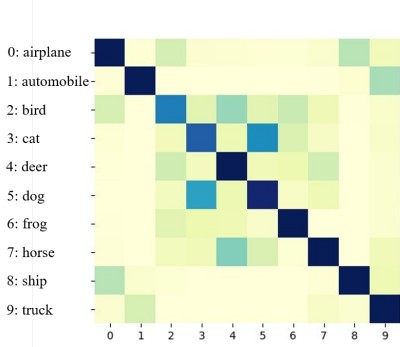

Figure A.3: Cosine similarity heat map of learned features from SimCLR on CIFAR-10 dataset. The darker the color, the larger the similarity.

## A  TECHNICAL DETAILS

### A.1  IMPLICIT BIAS OF SIMCLR ON CIFAR-10.

Figure A.3 plots the cosine similarity heat map of learned features from SimCLR on CIFAR-10 dataset. To calculate the similarity of class A (figures denoted by $a_i$) to class B (figures denoted by $b_i$), we first calculate the mean of $b_i$ as $\bar{b}$. Then, we sum up $\sum_i \text{sim}(a_i, \bar{b})$ and plot is with colors. Hence, the similarity matrix shown in Figure A.3 is not symmetric.

### A.2  PROOF OF PROPOSITION 3.2

Recall the domain-agnostic data augmentation process. For any $\boldsymbol{x}_i$, the probability density of having $\boldsymbol{t} \in \mathbb{R}^d$ as its augmented point can be characterized as

$$P_{\boldsymbol{t}|\boldsymbol{x}_i} = \mathbb{P}(\boldsymbol{x}_i \text{ and } \boldsymbol{x}_i' = \boldsymbol{t} \text{ form a positive pair } |\boldsymbol{x}_i) = \phi(\boldsymbol{t} - \boldsymbol{x}_i).$$

For isotropic Gaussian densities with mean 0 and covariance matrix $\sigma^2 \boldsymbol{I}$, $\phi(\boldsymbol{t} - \boldsymbol{x}_i) \propto \exp(-\|\boldsymbol{t} - \boldsymbol{x}_i\|_2^2 / 2\sigma^2)$, which is monotonic with the $l_2$ distance between $\boldsymbol{t}$ and $\boldsymbol{x}_i$.

### A.3  INVESTIGATIONS ON $C(f)$.

Figures A.4 and A.5 illustrate the evolution of different complexity measurements during the training process under the Gaussian mixture setting and the CIFAR-10 respectively.

In the Gaussian mixture setting, the feature extractor is a fully connected ReLU network. Besides $C(f)$, we also evaluate the popular sum of squared weights. The observations on SimCLR are listed as below:

- The expected Lipschitz constant $C(f)$ is small in initialization. It first increases (till around 100 iterations) and then consistently decreases. This empirically supports the implicit bias towards minimizing $C(f)$.
- $C(f)$ and the sum of squared weights share very similar patterns.
- The SNE loss is non-increasing, as if we are doing stochastic neighbor embedding using $l_2$-distance.

In the CIFAR-10 case, the feature extractor is ResNet-18 plus a fully-connected projection layer. The output from ResNet-18 is usually called representation (512 dimensional) and is utilized for downstream tasks while the projection (128 dimension) is used for training. Such a representation-projection set up is common in SSCL. Ma et al. (2023) aimed to decipher the projection head and revealed that the projection feature tends to be more uniformly distributed while the representation feature exhibits stronger alignment. Besides $C(f)$, we also evaluate the $l_2$-norm of the representation. The observations for SimCLR and $t$-SimCLR on CIFAR-10 are summarized as below:

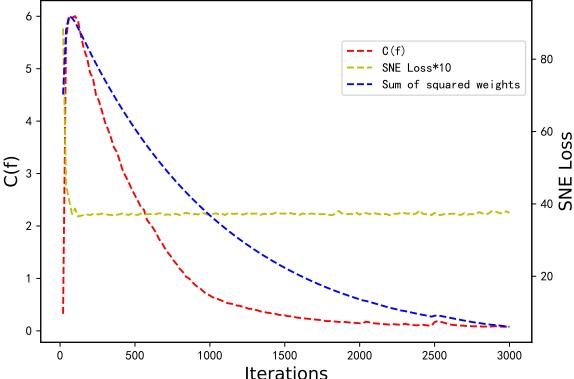

Figure A.4: Empirical evaluation on the complexity of the learned feature mapping during training under the Gaussian mixture setting. Two complexity measurements are considered, i.e., $C(f)$ as in (3.3) and the SNE loss as in (2.2). The SNE loss here only serves as in indicator for how well the pairwise distances are preserved. The training objective is the standard InfoNCE loss. The SNE loss decreases quickly until in the first 100 iterations and then stays flat.

- $C(f)$ for the projection layer shares similar patterns as in the Gaussian mixture case, first increase and then decreases. However, $C(f)$ for the representation layer monotonically decreases.

- $C(f)$ for the projection layer and the $l_2$-norm in the representation layer share almost identical patterns.

- Comparing SimCLR, both the the calculated $C(f)$ and $l_2$-norm are much smaller for $t$-SimCLR.

In conclusion, on one hand, our empirical results demonstrate that the complexity of the feature extractor $C(f)$ does decrease during training and seem to be implicitly minimized. On the other hand, its trend is shared with other more popularly used complexity measurements.

### A.4 PROOF OF COROLLARY 3.6

In this section, we illustrate with rigor how the hypothesized implicit bias can give rise to structure-preserving property of SSCL. Corollary 3.6 states that minimizing the (Lipschitz) complexity of the feature mapping will also result in the best match between $P$ and $Q$ (under permutation). To provide more theoretical insight, we present the following lemma in the simpler vector-matching case.

**Lemma A.1.** Let $0 < x_1 < \cdots < x_m$ and $0 < y_1 < \cdots < y_m$ be two real-valued sequences, normalized such that $\sum_{i=1}^m x_i^2 = \sum_{i=1}^m y_i^2 = 1$. Consider a permutation $\pi$ of $\{1, \cdots, m\}$ and denote all such permutations as $T$. Then

$$\operatorname*{argmin}_{\pi \in T} \sum_{i=1}^m \frac{y_{\pi(i)}}{x_i} = \operatorname*{argmin}_{\pi \in T} \sum_{i=1}^m (x_i - y_{\pi(i)})^2 := \pi^*,$$

where $\pi^*(i) = i$ for all $i = 1, \cdots, m$.

*Proof.* By the rearrangement inequality, we have

$$\sum_{i=1}^m \frac{y_{\pi(i)}}{x_i} \geq \sum_{i=1}^m \frac{y_i}{x_i}.$$

Similarly,

$$\sum_{i=1}^m (x_i - y_{\pi(i)})^2 = \sum_{i=1}^m x_i^2 + \sum_{i=1}^m y_i^2 - 2\sum_{i=1}^m x_i \cdot y_{\pi(i)} \geq 2 - 2\sum_{i=1}^m x_i \cdot y_i.$$

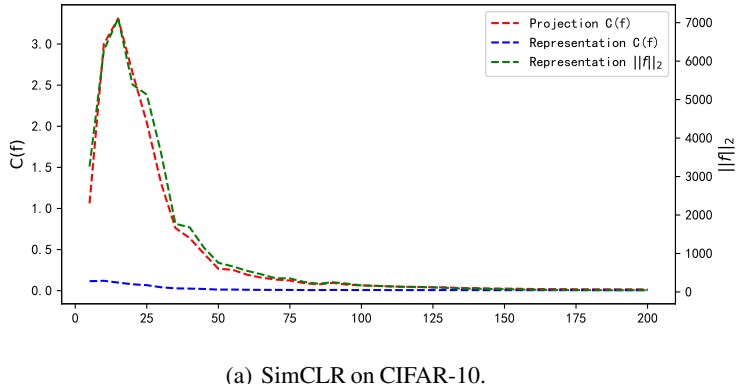

(a) SimCLR on CIFAR-10.

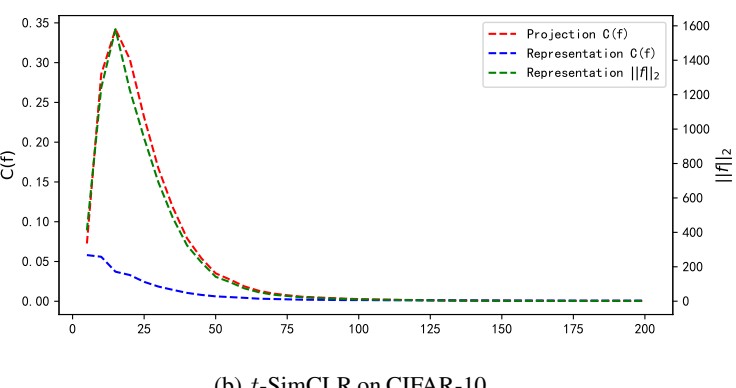

(b) $t$-SimCLR on CIFAR-10.

Figure A.5: Empirical evaluation on the complexity of the learned feature mapping during training on CIFAR-10. Two complexity measurements are considered, i.e., $C(f)$ as in (3.3) and $l_2$-norm. Specifically, we calculate the expected Lipschitz constant on both the representation layer (512-dimensional) and the projection layer (128-dimensional). Figure (a) and (b) show the trends (along the 200 training epochs) for SimCLR and $t$-SimCLR respectively.

$\square$

Lemma A.1 gives a vector-version illustration of our Corollary 3.6, stating that minimizing the expected derivative (to zero) of the mapping function $f$, i.e., $\sum_i f(x_i)/x_1$ leads to preserving the norm difference of the input vector and output vector.

Next, we provide the proof of Theorem 3.5.

*Proof of Theorem 3.5.* Straightforwardly, we can write

$$\|\bar{P} - Q^\pi\|_F = \sum_{i \neq j} \left( \frac{1}{p_{ij}} + q_{\pi(i)\pi(j)} \right)^2$$

$$= \sum_{i \neq j} \frac{1}{p_{ij}^2} + \sum_{i \neq j} q_{\pi(i)\pi(j)}{}^2 + 2 \sum_{i \neq j} \frac{q_{\pi(i)\pi(j)}}{p_{ij}}$$

$$= 2C_1(P, Q^\pi) + \sum_{i \neq j} \frac{1}{p_{ij}^2} + \sum_{i \neq j} q_{ij}{}^2$$

Thus, minimizing $C_1(P, Q^\pi)$ also minimizes $\|\bar{P} - Q^\pi\|_F$. $\square$

Theorem 3.5 is a straightforward generalization of Lemma A.1. Next, we provide proof for Corollary 3.6, restated below.

*Proof of Corollary 3.6.* Recall the SimCLR loss $L_{\text{InfoNCE}} = \frac{1}{2n} \sum_{i=1}^{n} (l(\boldsymbol{x}_i, \boldsymbol{x}_i') + l(\boldsymbol{x}_i', \boldsymbol{x}_i))$, where

$$l(\boldsymbol{x}_i, \boldsymbol{x}_i') = -\log \frac{\exp(\text{sim}(f(\boldsymbol{x}_i), f(\boldsymbol{x}_i'))/\tau)}{\sum_{x \in \mathcal{D}_n \cup \mathcal{D}_n' \setminus \{\boldsymbol{x}_i\}} \exp(\text{sim}(f(\boldsymbol{x}_i), f(\boldsymbol{x}))/\tau)}.$$

Without loss of generality, let $\tau = 1$. Notice that $l(\boldsymbol{x}_i, \boldsymbol{x}_i')$ is monotonically decreasing as $\text{sim}(f(\boldsymbol{x}_i), f(\boldsymbol{x}_i'))$ increases, due to the monotonicity of function $\frac{x}{x+c}$ with respect to $x > 0$ for any $c > 0$. Hence, in order for $L_{\text{InfoNCE}}$ to be minimized, perfect alignment is required, i.e., $f(\boldsymbol{x}_i) = f(\boldsymbol{x}_i')$ for any $i = 1, ..., n$.

With perfect alignment achieved, $L_{\text{InfoNCE}}$ only concerns the pairwise similarity between negative samples $f(\boldsymbol{x}_i)$'s, which can be simplified as $L_{\text{InfoNCE}} \geq L_{\text{uniform}}$ where

$$
\begin{aligned}
L_{\text{uniform}} &= \frac{1}{n} \sum_{i=1}^{n} -\log \frac{e}{e + \sum_{j \neq i} \exp(\text{sim}(f(\boldsymbol{x}_i), f(\boldsymbol{x}_j)))} \\
&\geq \log \left( \frac{1}{n} \sum_{i=1}^{n} \left( 1 + \frac{1}{e} \sum_{j \neq i} \exp(\text{sim}(f(\boldsymbol{x}_i), f(\boldsymbol{x}_j))) \right) \right) \\
&\geq \log \left( 1 + \frac{1}{n \cdot e} \sum_{1 \leq i \neq j \leq n} \exp(\text{sim}(f(\boldsymbol{x}_i), f(\boldsymbol{x}_j))) \right).
\end{aligned}
$$

$L_{\text{uniform}}$ can be minimized by mapping $\boldsymbol{x}_i$'s as distant as possible, hence the connection to Tammas problem and the uniformity principle.

With sufficient capacity of the feature mapping $f$, the SimCLR loss can be minimized to its (empirical) global minima. However, such $f$ is not unique since $L_{\text{InfoNCE}}$ is invariant to permutations of mapping relationships from $\boldsymbol{x}_i$ to $f(\boldsymbol{x}_i)$. If $f_n^*$ further minimizes $C(f)$ on the sample level, i.e.,

$$f_n^* := \underset{f}{\arg\min} \, C_n(f) = \underset{f}{\arg\min} \sum_{1 \leq i \neq j \leq n} \frac{\|f(\boldsymbol{x}_i) - f(\boldsymbol{x}_j)\|_2}{\|\boldsymbol{x}_i - \boldsymbol{x}_j\|_2},$$

Then, $f_n^*$ also solves a type of SNE problem with uniformity constraint (3.4) as stated in Theorem 3.5. To see this, if we define $q_{ij} = -\|f(\boldsymbol{x}_i) - f(\boldsymbol{x}_j)\|_2$ and $p_{ij} = -\|\boldsymbol{x}_i - \boldsymbol{x}_j\|_2$, which is reasonable since the larger the distance, the smaller the similarity, we can directly apply the results in Theorem 3.5.

$\square$

**Remark A.2.** As can be seen from Theorem 3.5 and the proof of Corollary 3.6, we showcase the relationship between minimizing $C(f)$ and structure preserving property by considering a special SNE problem, where the pairwise similarity is not modeled by Gaussian as standard, hence the word "resembling" in Corollary 3.6. Although $q_{ij} = -\|f(\boldsymbol{x}_i) - f(\boldsymbol{x}_j)\|_2$ is unorthodox, it is reasonable since the larger the distance, the smaller the similarity. If we consider the SNE method as in Hinton et al. (2006), our proof does not go through directly and demands more complicated analysis. However, our results are still valid in connecting the complexity of the feature map to the pairwise similarity matching.

Our statement in Corollary 3.6 requires perfect alignment or perfect uniformity. When the assumptions are not perfectly met, we can still obtain insights for the resulting feature mapping. Alignment and uniformity (Wang & Isola, 2020) is not the whole story of contrastive learning, and our identified structure-preserving property implicitly induced by complexity minimization provides an other angle of the learning process. From this perspective, contrastive learning can be thought of as a combination of alignment and SNE with uniformity constraint. In Figure A.3, while obtaining approximate alignment and uniformity, the feature mapping also preserves the relative relationships of the clusters (labels).

A.5 ALIGNMENT AND UNIFORMITY OF t-SIMCLR

Due to the change of training objective, we may want to reevaluate the properties of the learned feature from $t$-SimCLR. We will show that alignment still hold while uniformity is changed (to infinity).

Let us consider a compact region $\Omega \subset \mathbb{R}^d$ and $\boldsymbol{x}_i \in \Omega$. Let $t$ be the transformation such that the augmented data point $\boldsymbol{x}_i' = t(\boldsymbol{x}_i)$ is still in $\Omega$. Wang & Isola (2020) showed that the contrastive loss can be decomposed into the alignment loss and the uniformity loss. Zimmermann et al. (2021) further showed that the contrastive loss converges to the cross-entropy between latent distributions, where the underlying latent space is assumed to be uniform, and the positive pairs are specified to be an exponential distribution. In this section, we show a parallel result, which states that in the population level, the $t$-SNE loss is the cross-entropy between two distributions of generating positive pairs.

**Theorem A.3.** Let $H(\cdot,\cdot)$ be the cross entropy between distributions. Let $p(\boldsymbol{x})$ be the density of $\boldsymbol{x}$, $p(\cdot|\boldsymbol{x})$ be the conditional density of generating a positive pair, and define

$$q_f(\boldsymbol{x}'|\boldsymbol{x}) = C_f(\boldsymbol{x})^{-1} \frac{p(\boldsymbol{x}')}{1 + \|f(\boldsymbol{x}) - f(\boldsymbol{x}')\|_2^2}, \text{ with } C_f(\boldsymbol{x}) = \int_\Omega \frac{p(\boldsymbol{x}')}{1 + \|f(\boldsymbol{x}) - f(\boldsymbol{x}')\|_2^2} \, \mathrm{d}\boldsymbol{x}'.$$

Then, we have

$$\mathbb{E}_{\boldsymbol{x} \sim p(\boldsymbol{x})}(H(p(\cdot|\boldsymbol{x}), q_f(\cdot|\boldsymbol{x})) = L_a(f) + L_u(f), \tag{A.1}$$

which corresponds to the population-level $t$-SimCLR loss where

$$L_a = \mathbb{E}_{\boldsymbol{x} \sim p(\boldsymbol{x})} \mathbb{E}_{\boldsymbol{x} \sim p(\boldsymbol{x}'|\boldsymbol{x})} \log(1 + \|f(\boldsymbol{x}) - f(\boldsymbol{x}')\|_2^2),$$
$$L_u = \mathbb{E}_{\boldsymbol{x} \sim p(\boldsymbol{x})} \log \mathbb{E}_{\widetilde{\boldsymbol{x}} \sim p(\widetilde{\boldsymbol{x}})} (1 + \|f(\boldsymbol{x}) - f(\widetilde{\boldsymbol{x}})\|_2^2)^{-1}.$$

*Proof.* Note that

$$H(p(\cdot|\boldsymbol{x}), q_f(\cdot|\boldsymbol{x}))$$

$$= -\int_\Omega p(\boldsymbol{x}'|\boldsymbol{x}) \log\left(\frac{p(\boldsymbol{x}')}{1 + \|f(\boldsymbol{x}) - f(\boldsymbol{x}')\|_2^2}\right) \mathrm{d}\boldsymbol{x}' + \log C_f(\boldsymbol{x})$$

$$= \int_\Omega p(\boldsymbol{x}'|\boldsymbol{x}) \log(1 + \|f(\boldsymbol{x}) - f(\boldsymbol{x}')\|_2^2) \mathrm{d}\boldsymbol{x}' - \int_\Omega p(\boldsymbol{x}'|\boldsymbol{x}) \log(p(\boldsymbol{x}')) \mathrm{d}\boldsymbol{x}' + \log \int_\Omega \frac{p(\boldsymbol{x}')}{1 + \|f(\boldsymbol{x}) - f(\boldsymbol{x}')\|_2^2} \mathrm{d}\boldsymbol{x}'$$

$$= \int_\Omega p(\boldsymbol{x}'|\boldsymbol{x}) \log(1 + \|f(\boldsymbol{x}) - f(\boldsymbol{x}')\|_2^2) \mathrm{d}\boldsymbol{x}' - \int_\Omega p(\boldsymbol{x}'|\boldsymbol{x}) \log(p(\boldsymbol{x}')) \mathrm{d}\boldsymbol{x}' + \log \mathbb{E}_{\boldsymbol{x}' \sim p(\boldsymbol{x}')} (1 + \|f(\boldsymbol{x}) - f(\boldsymbol{x}')\|_2^2)^{-1}.$$

Taking expectation with respect to $\boldsymbol{x}$ leads to

$$\mathbb{E}_{\boldsymbol{x} \sim p(\boldsymbol{x})} H(p(\cdot|\boldsymbol{x}), q_f(\cdot|\boldsymbol{x}))$$

$$= \mathbb{E}_{\boldsymbol{x} \sim p(\boldsymbol{x})} \mathbb{E}_{\boldsymbol{x}' \sim p(\boldsymbol{x}'|\boldsymbol{x})} \log(1 + \|f(\boldsymbol{x}) - f(\boldsymbol{x}')\|_2^2) + \mathbb{E}_{\boldsymbol{x} \sim p(\boldsymbol{x})} \log \mathbb{E}_{\widetilde{\boldsymbol{x}} \sim p(\widetilde{\boldsymbol{x}})} (1 + \|f(\boldsymbol{x}) - f(\widetilde{\boldsymbol{x}})\|_2^2)^{-1}$$

$$- \int_\Omega \int_\Omega p(\boldsymbol{x}) p(\boldsymbol{x}'|\boldsymbol{x}) \log(p(\boldsymbol{x}')) \mathrm{d}\boldsymbol{x}' \mathrm{d}\boldsymbol{x}$$

$$= L_a(f) + L_u(f) - C_p,$$

where

$$C_p = \int_\Omega \int_\Omega p(\boldsymbol{x}) p(\boldsymbol{x}'|\boldsymbol{x}) \log(p(\boldsymbol{x}')) \mathrm{d}\boldsymbol{x}' \mathrm{d}\boldsymbol{x} = \int_\Omega \int_\Omega p(\boldsymbol{x}, \boldsymbol{x}') \log(p(\boldsymbol{x}')) \mathrm{d}\boldsymbol{x}' \mathrm{d}\boldsymbol{x}$$

does not depend on $f$.

$$\mathbb{E}_{\boldsymbol{x} \sim p(\boldsymbol{x})} H(p(\cdot|\boldsymbol{x}), q_f(\cdot|\boldsymbol{x}))$$

$$= \int_\Omega p(\boldsymbol{x}) \frac{1}{p(\boldsymbol{x})} \int_\Omega p(\boldsymbol{x}, \boldsymbol{x}') \log\left(\frac{p(\boldsymbol{x}')}{1 + \|f(\boldsymbol{x}) - f(\boldsymbol{x}')\|_2^2}\right) \mathrm{d}\boldsymbol{x}' \mathrm{d}\boldsymbol{x}$$

$$- \int_\Omega \int_\Omega \frac{p(\boldsymbol{x}) p(\boldsymbol{x}')}{1 + \|f(\boldsymbol{x}) - f(\boldsymbol{x}')\|_2^2} \mathrm{d}\boldsymbol{x} \mathrm{d}\boldsymbol{x}'$$

$$= \int_\Omega \int_\Omega p(\boldsymbol{x}, \boldsymbol{x}') \log\left(\frac{p(\boldsymbol{x}')}{1 + \|f(\boldsymbol{x}) - f(\boldsymbol{x}')\|_2^2}\right) \mathrm{d}\boldsymbol{x}' \mathrm{d}\boldsymbol{x}$$

$$- \int_\Omega \int_\Omega \frac{p(\boldsymbol{x}) p(\boldsymbol{x}')}{1 + \|f(\boldsymbol{x}) - f(\boldsymbol{x}')\|_2^2} \mathrm{d}\boldsymbol{x} \mathrm{d}\boldsymbol{x}'.$$

$\square$

In Theorem A.3, $L_a$ is the alignment loss and $L_u$ is the uniformity loss. The decomposition is much more natural for $t$-SimCLR as opposed to that in $L_{\text{InfoNCE}}$, mainly due to the change from conditional to joint distribution when modeling the pairwise similarity. Furthermore, if the $t$-SimCLR loss is minimized, we must have $p(\cdot|\boldsymbol{x}) = q_f(\cdot|\boldsymbol{x})$, provided $f$ has sufficient capacity. Note that if $p(\cdot|\boldsymbol{x}) = q_f(\cdot|\boldsymbol{x})$, then $P_{j|i}$ and $Q_{j|i}$ are perfectly matched, which indicates that we obtain a perfect neighbor embedding.

Theorem A.3 implies that the optimal feature mapping $f^*$ satisfies
$$p(\cdot|\boldsymbol{x}) = q_{f^*}(\cdot|\boldsymbol{x}),$$
which further implies that for any $\boldsymbol{x} \in \Omega$,
$$C_{f^*}(\boldsymbol{x})^{-1} \frac{p(\boldsymbol{x}')}{1 + \|f^*(\boldsymbol{x}) - f^*(\boldsymbol{x}')\|_2^2} \propto C(\boldsymbol{x})^{-1} p(\boldsymbol{x}'|\boldsymbol{x})$$
$$\Leftrightarrow C_{f^*}(\boldsymbol{x})^{-1} \frac{1}{1 + \|f^*(\boldsymbol{x}) - f^*(\boldsymbol{x}')\|_2^2} \propto C(\boldsymbol{x})^{-1} \frac{p(\boldsymbol{x}, \boldsymbol{x}')}{p(\boldsymbol{x})p(\boldsymbol{x}')}, \tag{A.2}$$
where $C(\boldsymbol{x}) = \int p(\boldsymbol{x}'|\boldsymbol{x}) \mathrm{d}\boldsymbol{x}'$. Unlike the usual normalized SimCLR, $t$-SNE does not assume any special structure on $f$ (e.g., $\|f\|_2 = 1$), thus $f$ can go to infinity. Comparing to the finite sample $t$-SimCLR loss, the population version is trickier to analyze. This is because for a given point $\boldsymbol{x}'$, it can be an augmented sample of some $\boldsymbol{x}$ (with probability $p(\boldsymbol{x}'|\boldsymbol{x})$), or a negative sample of $\boldsymbol{x}$ (when we treat $\boldsymbol{x}'$ as another sample point). This reflects the essential difficulty between population and finite samples in contrastive learning, not only for $t$-SimCLR.

For clustered data, (A.2) provides two important messages, provided that the augmentation is not too extreme and the augmented sample $\boldsymbol{x}'$ stays in the same cluster as the original $\boldsymbol{x}$. On one hand, when $\boldsymbol{x}_1$ and $\boldsymbol{x}_2$ belongs to different clusters, the joint density $p(\boldsymbol{x} = \boldsymbol{x}_1, \boldsymbol{x}' = \boldsymbol{x}_2)$ will be very small, close to zero, which indicates that $\|f^*(\boldsymbol{x}_1) - f^*(\boldsymbol{x}_2)\|_2$ is very large, tending to infinity. On the other hand, for $\boldsymbol{x}_1$ and $\boldsymbol{x}_2$ belonging to the same cluster, $p(\boldsymbol{x} = \boldsymbol{x}_1, \boldsymbol{x}' = \boldsymbol{x}_2)$ will be relatively large. Hence, the features of the same cluster will stay close. Overall, we will observe similar clustered structure in the feature space. This is confirmed in the Gaussian mixture setting in Figure 1(c), in which case, the problem can be oversimplified as mapping 5 points in $\mathbb{R}^2$ to the unit-circle.

## B    CONNECTION TO DISTANCE BETWEEN DISTRIBUTIONS

Through the lens of stochastic neighbor embedding, the feature learning process of SSCL methods can be seen as minimizing certain "distances" between distributions in *different dimensions*. Ideally, the feature should preserve the distributional information about the data. Since the data and the feature do not lie in the same metric space, quantitatively measuring their distributional distance is difficult. Fortunately, there are existing tools we can utilize, specifically, Gromov-Wasserstein distance (Mémoli, 2011; Salmona et al., 2021).

Let $\mathcal{X}$, $\mathcal{Z}$ be two Polish spaces, each endowed respectively with probability measures $p_x$ and $p_z$. Given two measurable cost functions $c_x : \mathcal{X} \times \mathcal{X} \to \mathbb{R}$, $c_z : \mathcal{Z} \times \mathcal{Z} \to \mathbb{R}$, and $D : \mathbb{R} \times \mathbb{R} \to \mathbb{R}$, the Gromov-Wasserstein distance can be defined as
$$GW_p(p_x, p_z | c_x, c_z) := \left( \inf_{\pi \in \prod(p_x, p_z)} \int_{\mathcal{X}^2 \times \mathcal{Z}^2} D(c_x(x, x'), c_z(z, z'))^p d\pi(x, z) d\pi(x', z') \right)^{1/p},$$
where $\prod(p_x, p_z)$ denotes all the joint distributions in $\mathcal{X} \times \mathcal{Z}$ such that the marginals are $p_x$ and $p_z$. Typically, $D(c_x, c_z)$ is chosen to be $|c_x - c_z|$ and $c_x(x, x')$ is usually chosen to be $\|x - x'\|_p$. The key idea of the Gromov-Wasserstein distance to circumvent the dimension mismatch is to change from comparing marginal distribution to pairwise distributions, which is very similar to the SNE objective. Consider Monge's formulation of the optimal transportation problem and let $z = f(x)$. By choosing $c_z(z_i, z_j) = \log(\widetilde{Q}_{j|i})$ with $\widetilde{Q}$ specified as in (3.1), $c_x(x_i, x_j) = P_{j|i}$ with $\widetilde{P}$ specified as in (3.2) and letting $D(c_x, c_z) = c_x(\log(c_x) - \log(c_z))$, we have
$$GW_1(p_x, p_{f(x)}) \le \mathbb{E}_{x, x'}(D(c_x(x, x'), c_z(f(x), f(x')))),$$
where the right hand side recovers the expected InfoNCE loss. Hence, the SNE perspective can also be viewed as minimizing the Gromov-Wasserstein distance between $p_z$ and $p_x$.

It is worth noting that such an interpretation only relates to contrastive learning, not including generative-based self-supervised learning methods such as Masked AutoEncoder (MAE) (He et al., 2021).

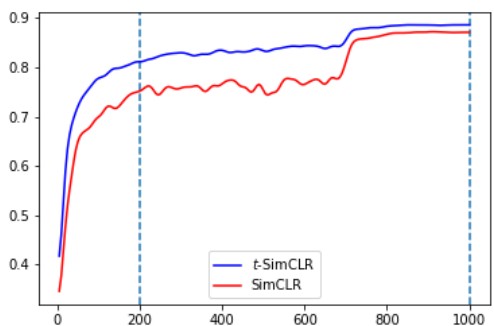

Figure C.6: Nearest neighbor test accuracy vs. training epochs. SimCLR and $t$-SimCLR share similar trends and convergence speed.

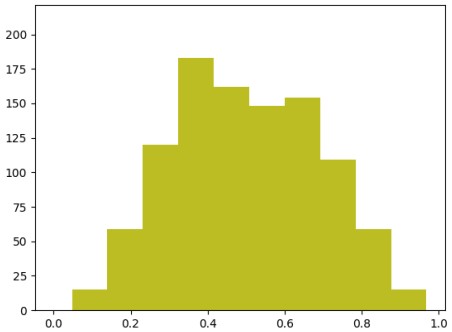

Figure C.7: The histogram of IoUs for 1000 constructed positive pairs in CIFAR-10. The empirical distribution is almost symmetric around 0.5.

## C    EXPERIMENT DETAILS

### C.1    CIFAR-10 SETTINGS

CIFAR-10 (Krizhevsky, 2009) is a colorful image dataset with 50000 training samples and 10000 test samples from 10 categories. We use ResNet-18 (He et al., 2016) as the feature extractor, and the other settings such as projection head all follow the original settings of SimCLR (Chen et al., 2020a). To evaluate the quality of the features, we follow the KNN evaluation protocol (Wu et al., 2018). which computes the cosine similarities in the embedding space between the test image and its nearest neighbors, and make the prediction via weighted voting. We train each model with batch size of 256 and 200 epochs for quicker evaluation. For $t$-SimCLR, without specifying otherwise, we grid search the $t_{df}$ and $\tau$ with range $\{1, 2, 5, 10\}$ and $\{1, 2, 5, 10\}$ respectively.

**Ablation of training epochs**    We also run the SimCLR and $t$-SimCLR experiments in the more standard 1000 epochs setting. For SimCLR, we use batch size of 512, learning rate of 0.3, temperature of 0.7, and weight dacay of 0.0001. For $t$-SimCLR, we use batch size of 512, learning rate of 0.8, temperature of 10, weight dacay of 0.0002, and $t_{df} = 5$. The nearest neighbor accuracy for SimCLR is 87.2% vs. that for $t$-SimCLR is 88.8%.

### C.2    IMAGE AUGMENTATION

When processing images, several popular augmentations are usually adopted (following the setting in SimCLR Chen et al. (2020a)), e.g., random resized crop (crops a random portion of image and resize it

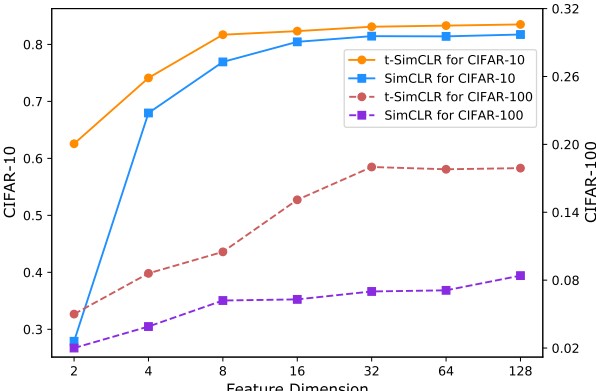

Figure C.8: Extension on Figure 2(b). Nearest neighbor classification accuracy for SimCLR vs. $t$-SimCLR on both CIFAR-10 (in-distribution) and CIFAR-100 (out-of-distribution) using different feature dimensions.

to the original size), horizontal flip, color jitter (randomly change the brightness, contrast, saturation and hue of an image). To illustrate the natural weighting scheme in Section 4.1, we considered random re-sized crop and specifies the weights by the IoU (intersection over union) of the positive pair. In particular, two augmented images are created from an anchor image. Each augmentation crops a rectangular region of the image, denoted by $r_1, r_2$ respectively, and their IoU is defined by the area of intersection $r_1 \cap r_2$ divided by the area of the union $r_1 \cup r_2$. The IoU is always between 0 and 1. In our experiment, we chose the default settings and Figure C.7 illustrates the IoU histogram of 1000 constructed positive pairs.

### C.3  DEGREE OF FREEDOM IN $t$-SIMCLR

**Feature dimension efficiency in OOD case.**  To further investigate the generalization ability of SSCL methods, we devise a challenging setting where the model is trained on CIFAR-10 and tested on CIFAR-100 classification. In this case, we evaluate the effect of increasing feature dimensions in the projection layer, as an extension on the CIFAR-10 in-distribution case. The results are shown in Figure C.8, where there are two things to note:

- The gain of extra dimensions in the OOD case does vanish later than that in the in-distribution case.
- The advantage of SimCLR vs. $t$-SimCLR is very significant with around 10% improvement when $d = 128$ using nearest neighbor[5] classification, indicating that $t$-SimCLR produces better separated clusters.

**Relationship between $t_{df}$ and $d_z$.**  The larger the degree of freedom $t_{df}$, the less heavy-tail the t-distribution. As $d_z$ decreases, the crowding problem becomes more severe and as recommended by (Van der Maaten & Hinton, 2008), a smaller $t_{df}$ tends to work better. We evaluate the sensitivity of $t_{df}$ (1, 5, 10) under different choices of $d_z$ (1, 2, 4, 8, 16, 32, 64, 128) in CIFAR-10 and the results are reported in Figure C.9. As can be seen, when $d_z$ is small (1,2,4,8), $t_{df} = 1$ outperforms. Comparing $t_{df} = 5$ and $t_{df} = 10$, the two perform similarly when $d_z$ is large (16,32,64,128) but the smaller $t_{df} = 5$ yields better accuracy when $d_z = 1,2,4$.

**Tuning temperature vs. tuning $t_{df}$.**  As illustrated in Section 4.2, when the feature space dimension is low, the heavy-tailed t-distribution is a better choice than Gaussian to alleviate the crowding problem.

---

[5]When evaluating by training linear classifiers for 100 epochs, the accuracy for SimCLR is 46.4% and that for $t$-SimCLR is 48.14% (averaged over 3 replications).

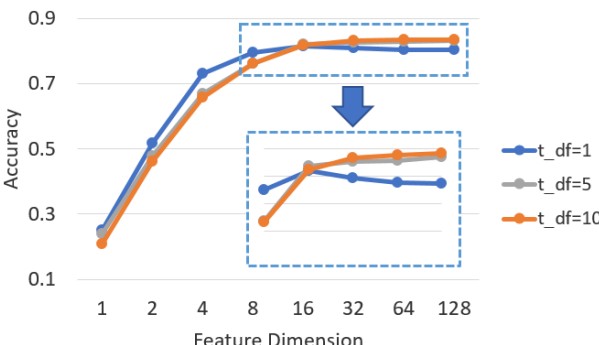

Figure C.9: Nearest neighbor classification accuracy on CIFAR-10 for $t$-SimCLR using different feature dimensions and different degrees of freedom (t_df).

Even though tuning the temperature of $L_{\mathrm{InfoNCE}}$, i.e., making $\tau$ larger, can also have the effect of making the distribution less concentrated ($\tau$ can be seen as the standard deviation), tuning temperature and tuning $t_{df}$ are fundamentally different. The former is controlling how fast does the similarity $Q_{i,j}$ decays as the distance between $z_i$ and $z_j$ increases, while the latter serves as a scaling factor, offering constant level modification of the scheme. In our experiments with SimCLR vs $t$-SimCLR on CIFAR-10, temperature is tuned as a hyperparameter. The difference in $\tau$ can never make up to the difference between the baseline SimCLR and $t$-SimCLR. We found $\tau = 0.5$ to work better for the base SimCLR while larger $\tau$ works better with our $t$-SimCLR. We recommend $\tau = 5$ as the default choice.

### C.4 IMAGENET PRE-TRAINING

To show the ability for large scale domain transfer and OOD generalization, we conduct experiments on ImageNet pre-training based on MoCo-v2 with its official implementation[6]. We follow most of their settings, e.g, data augmentation, 200 epochs pre-training, and optimization strategy, etc. The loss is modified according to Section 4.2 and batch normalization is applied along every dimension. We grid search the $t_{df}$ and $\tau$ with range $\{2, 5, 10, 15\}$ and $\{0.2, 2, 5, 10\}$ respectively. Finally we choose $t_{df} = 10$ and $\tau = 5$ to be the optimal hyperparameters. We use this pre-train model as initialization for domain transfer and OOD experiments.

### C.5 DOMAIN TRANSFER

We compare MoCo-v2 pre-trained with 800 / 200 epochs and $t$-MoCo-v2 on Aircraft, Birdsnap, Caltech101, Cars, CIFAR10, CIFAR100, DTD, Pets, and SUN397 in Table C.3. We follow the transfer settings in Ericsson et al. (2021) to finetune the pre-trained models. For datasets Birdsnap, Cars, CIFAR10, CIFAR100, DTD, and SUN397, we report the top-1 accuracy metric, while for Aircraft, Caltech101, and Pets, we report the mean per-class accuracy metric. We also follow Ericsson et al. (2021) to split each dataset into training, validation, and test sets. On each dataset, we perform a hyperparameter search as follows. (1) We choose the initial learning rate according to a grid of 4 logarithmically spaced values between $1 \times 10^{-4}$ and $1 \times 10^{-1}$; (2) We choose the weight decay parameter according to a grid of 4 logarithmically spaced values between $1 \times 10^{-6}$ and $1 \times 10^{-3}$, plus no weight decay; (3) The weight decay values are divided by the learning rate; (4) For each pair of learning rate and weight decay, we finetune the pre-trained model for 5000 steps by SGD with Nesterov momentum 0.9, batch size of 64, and cosine annealing learning rate schedule without restarts. As can be seen in Table C.3, our $t$-MoCo-v2 with 200 epochs even outperform the baseline with 800 epochs on average.

### C.6 OOD GENERALIZATION

To demonstrate the advantage of our modification, we also compare MoCo-v2 pre-trained with 800 / 200 epochs and $t$-MoCo-v2 on OOD generalization benchmarks: PACS Li et al. (2017), VLCS Fang et al. (2013), Office-Home Venkateswara et al. (2017). We follow the standard way to conduct the

---

[6]https://github.com/facebookresearch/moco

Table C.3: Domain transfer results of vanilla MoCo-v2 and $t$-MoCo-v2.

| Method | Aircraft | Birdsnap | Caltech101 | Cars | CIFAR10 | CIFAR100 | DTD | Pets | SUN397 | Avg. |
|---|---|---|---|---|---|---|---|---|---|---|
| MoCo-v2 (800 epochs) | **83.80** | 45.51 | 83.01 | **86.18** | **96.42** | 71.69 | **71.70** | **89.11** | 55.61 | 75.89 |
| MoCo-v2 (200 epochs) | 82.75 | 44.53 | 83.31 | 85.24 | 95.81 | 72.75 | 71.22 | 86.70 | 56.05 | 75.37 |
| $t$-MoCo-v2 (200 epochs) | 82.78 | **53.46** | **86.81** | 86.17 | 96.04 | **78.32** | 69.20 | 87.95 | **59.30** | **77.78** |

Table C.4: OOD accuracies of vanilla MoCo-v2 and $t$-MoCo-v2 on domain generalization benchmarks.

| Method | PACS | VLCS | Office-Home | Avg. |
|---|---|---|---|---|
| MoCo-v2 (800 epochs) | 58.9 | 69.8 | 41.6 | 56.8 |
| MoCo-v2 (200 epochs) | 58.5 | 70.4 | 36.6 | 55.2 |
| $t$-MoCo-v2 (200 epochs) | **61.3** | **75.1** | **42.1** | **59.5** |

experiments, i.e., choosing one domain as the test domain and using the remaining domains as training domains, which is named the leave-one-domain-out protocol. The top linear classifier is trained on the training domains and tested on the test domain. Each domain rotates as the test domain and the average accuracy is reported for each dataset in Table C.4. On each dataset, we perform a hyperparameter search following DomainBed Gulrajani & Lopez-Paz (2021). We adopt the leave-one-domain-out cross-validation setup in DomainBed with 10 experiments for hyperparameter selection and run 3 trials. As can be seen in Table C.4, our $t$-MoCo-v2 with 200 epochs even significantly outperform the baseline with 800 epochs for all of the three datasets.

## C.7 SSCL INSPIRED DATA VISUALIZATION

$t$-SNE (Van der Maaten & Hinton, 2008) and its variants are designed for data visualization. However, for more complicated data, such as colored images, the results are not satisfactory. Using standard $t$-SNE, the 2D visualization of the 50K training images of CIFAR-10 (labels denoted as 0, 1,...,9) can be seen in Figure C.10, where different labels are hardly separated. The poor performance of $t$-SNE on CIFAR-10 can be traced back to the poor distance choice on images, i.e., $l_2$-norm. Inspired by the success of SSCL for natural images, $t$-SNE can potentially be improved by incorporating data augmentations.

In light of our perspective (S1), $t$-SNE can take advantage of the distance specified with (3.2) and the resulting model is essentially our $t$-SimCLR with feature dimension 2. The visualization from $t$-SimCLR is shown in Figure C.11, which is much more separated (the nearest neighbor classification accuracy on CIFAR-10 test data is 56.6%). By choosing the feature dimension to be 2, various SSCL methods can also be made into data visualizing tools. In Figure C.12, we visualize the outcome from SimCLR (the nearest neighbor classification accuracy on CIFAR-10 test data is 24.8%).

Similar investigations have been carried in Böhm et al. (2022); Damrich et al. (2022) where they focused specifically on data visualization and stochastic neighbor embedding.

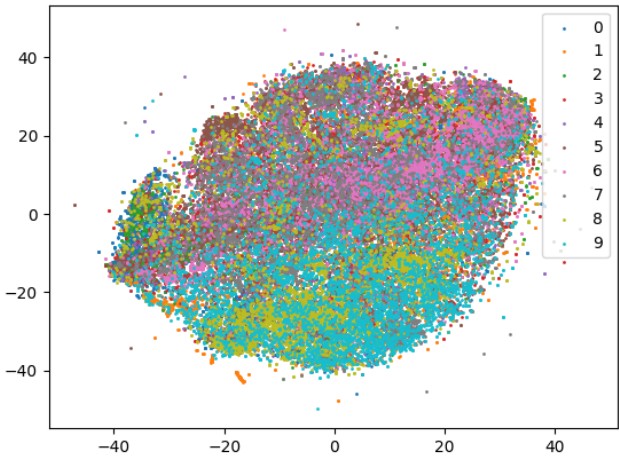

Figure C.10: 50K CIFAR-10 training images visualization in 2D with $t$-**SNE**.

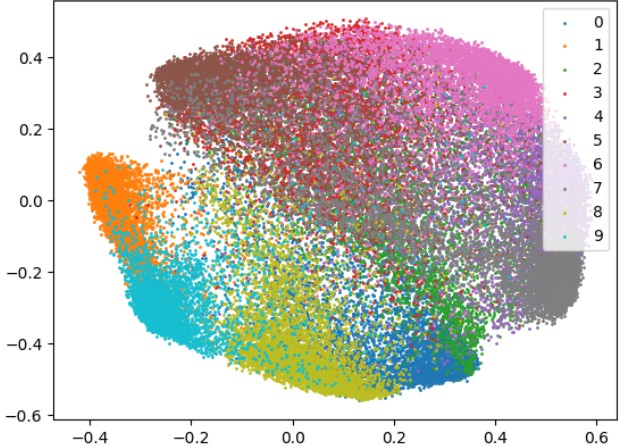

Figure C.11: 50K CIFAR-10 training images visualization in 2D with the default $t$-**SimCLR**.

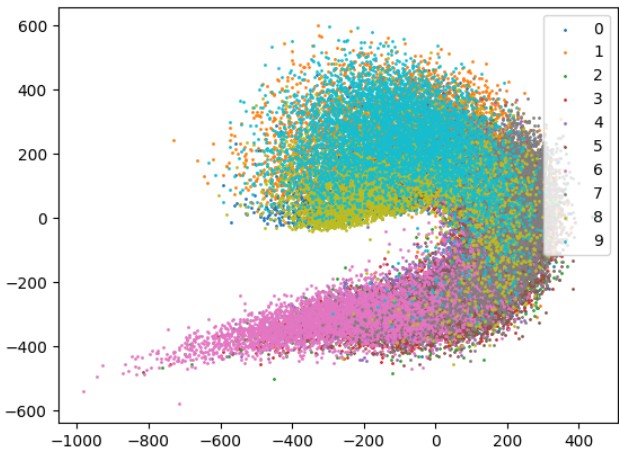

Figure C.12: 50K CIFAR-10 training images visualization in 2D with the **SimCLR**.

