# OpenReview forum: "Your Contrastive Learning Is Secretly Doing Stochastic Neighbor Embedding"
_ICLR.cc/2023/Conference — ICLR 2023 poster_

### Official Review · Reviewer_qnUF · 2022-10-13

**Confidence:** 4
**Correctness:** 3
**Technical Novelty And Significance:** 2
**Empirical Novelty And Significance:** 2
**Recommendation:** 6

**Clarity, Quality, Novelty And Reproducibility:**

Clarity:
In general, this paper is written well. Some of the theoretical claims need to be more precise. For example, in Proposition 3.2, what's the data set and how are the positive samples defined?

Quality:
I think the SNE perspective of SSCL is interesting and the empirical results look promising. However, I think the theoretical analysis in this paper is too restricted and lacks depth.

Novelty:
As far as I know, this is the first paper that connects SSCL and SNE.

Reproducibility:
I think the theoretical results and experiments in this paper are reproducible.

**Strength And Weaknesses:**

Strengths:
I think viewing SSCL as a type of SNE is an exciting and potentially fruitful perspective. Many theoretical analyses and practical techniques can be potentially applied to SSCL to improve its performance. As already demonstrated in this paper, the re-weighting of positive pairs and t-SNE style matching inspired by SNE literature can improve the performance of SSCL.

Weaknesses:
The equivalence between SSCL and SNE is not hard to see after a re-writing of the objective function; this connection is interesting but not very surprising. The bulk of this paper focuses on the theoretical analysis of SSCL from the SNE perspective. However, all the theoretical analysis is restricted to toy examples and some arguments are handwavy and lack depth.
1. The analysis is restricted to a toy example: the gaussian mixture model. The positive samples are generated by sampling from the same component. This is impossible to conduct in SSCL since we cannot access the label information in SSCL and therefore cannot decide which component a sample belongs to.
2. In the analysis domain-agnostic data augmentation, why is $P(x_1 \text{ and } x_2 \text{ form a positive pair })$ equal to the density $\phi(x_1-x_2)$? For example, suppose the original dataset only contains one sample, all the augmented data are positive pairs, in which case $P(x_1 \text{ and } x_2 \text{ form a positive pair })$ always equals $1$ but the density may not be $1$.
3. Regarding the implicit bias, this paper hypothesized that SSCL minimizes the expected Lipschitz constant $C(f)$. Then it proves that minimizing $C(f)$ in SSCL is equivalent to solving an SNE problem with uniformity constraint. But the SNE problem is not standard ($P$ needs to be transformed in a strange way and $Q$ is also unorthodox). This paper did not prove or give sufficient empirical evidence to show SimCLR is minimizing $C(f)$.
4. The arguments for the effects of normalization on OOD generalization are very handwavy and the authors did not provide any rigorous theorem for this part.


**Summary Of The Paper:**

1. This paper interpreted self-supervised contrastive learning (SSCL) as a type of stochastic neighbor embedding (SNE) methods that preserve the pairwise similarities specified by the data augmentations.
2. Based on the connection with SNE, this paper provided theoretical insights for domain-agnostic data augmentation, implicit bias, and OOD generalization of SSCL. In particular, for the implicit bias of SSCL, this paper proved the equivalence between minimizing the expected Lipschitz constant of the SSCL feature map and SNE with uniformity constraint.
3. Motivated by the SNE perspective, this paper proposed modifications to the SSCL methods, including re-weighting positive pairs and t-SNE style matching, and achieved improvements in experiments.

**Summary Of The Review:**

I think building the connection between SSCL and SNE is very important and many techniques in SNE can be applied to SSCL to improve its performance (as already demonstrated in this paper).

My major concern with this paper is the theoretical part. I found the theoretical analysis in this paper restricted to exceedingly simple toy examples and some arguments are handwavy and lack depth. The connection between the implicit bias of SSCL (minimizing $C(f)$) and SNE with uniformity constraint is pretty interesting. Unfortunately, this paper fails to either prove or empirically demonstrate that SimCLR is minimizing $C(f)$.

---

> ### Author Response · Authors · 2022-11-12
> **Response to Reviewer qnUF (part 1)**
>
> Thank you very much for the constructive criticism.
>
> The main goal of this work is to provide new insights of SSCL from the perspective of SNE.
> When presenting, we often start by showing insights in the simple Gaussian mixture setting, and demonstrate that the same insights can be carried over to real applications, e.g., image classification, by empirical evaluation.
> We agree with you that our work can definitely benefit from deeper and more concrete theoretical analysis. For instance, bridging the gap between the simple Gaussian mixture setting and real image data would certainly be ideal, but it incurs enormous technical difficulty and we will leave them for future works. We sincerely hope our SNE perspective can inspire more investigations along this direction.
>
> Below we address the raised weaknesses separately:
>
> ### 1.
> The Gaussian case mainly serves as a proof of concept for intuitive illustrations. Though over-simplified, we do believe that it is reasonable and relevant. However, it should be noted that our analysis is general and does not rely on the Gaussian mixture setting. We have added clarifications on this at the beginning of Sec 3.1.
>
> Next, we clarify the positive pairs of the Gaussian mixture setting.
> The key idea of constructing positive pairs is to create the so-called different views of the same image. Different views ideally share the same semantic meaning (label). In the CIFAR-10 case, it is reasonable to believe that there are 10 well-separated clusters for the underlying distribution, each corresponding to one of the labels. We also believe that popular data augmentation techniques including crop and flip do not change a cat into a dog, though we do not have access to the label information.
> In order to mimic such augmentations in real data, in the toy Gaussian mixture setting, we also generate the positive sample pair from the same component. It should be noted that the label of the component is not important in generating the positive samples, as long as they are from the same cluster or component.
> Another possible augmentation in the toy case can be domain-agnostic, e.g., adding Gaussian noises, which does not require the label information. However, we believe the former setting relates better to practice.
>
>
> ### 2.
> We apologize for not being more clear.
> In our case, we do not consider any training data $x_1, ..., x_n$.
> In the original statement, $x_1$ and $x_2$ are used to denote two arbitrary points in the input space.
> $P(x_1$ and $x_2$ form a positive pair $)$ is better understood as given $x_1$ (or $x_2$), the probability of constructing $x_2$ (or $x_1$) as its positive pair, or vice versa.
> Hence, given $x_1$ (or $x_2$), $P(x_1$ and $x_2$ form a positive pair $)$ is equal to the probability of sampling $\delta = x_2-x_1$ (or $\delta = x_1-x_2$).
> We have changed the notation and rewritten the paragraph in the updated draft to make the argument more clear.
>
>
>
> ### 3.
> The SNE problem is indeed not standard. The monotonic transform is for technical reasons.
> SimCLR has the implicit tendency towards minimizing $C(f)$ is not proven in this work.
> We have added new experiments investigating the complexity $C(f)$ during the training process, in both the Gaussian mixture setting and the CIFAR-10 setting.
> The results are shown in the added Section A.2, with added Figures A.4 and A.5. Also see Overall Response 1.1 for more details.
>
> The findings can be summarized as:
> * The complexity of the feature extractor $C(f)$ does decrease during training and seems to be implicitly minimized.
>
> * On the other hand, its trend is shared with other more popularly used complexity measurements, e.g., $l_2$-norm of the features.
> There exists fruitful literature on characterizing implicit bias of training overparameterized neural networks as minimizing functional norms, e.g., $l_2$-norms [1,2] and RKHS norms [3,4]. Our expected Lipschitz constant $C(f)$ may seem less common, but we believe it is also being minimized implicitly during training.
>
> Reference:
>
> [1] Gunasekar, Suriya, et al. "Implicit bias of gradient descent on linear convolutional networks." Advances in Neural Information Processing Systems 31 (2018).
>
> [2] Gunasekar, Suriya, et al. "Characterizing implicit bias in terms of optimization geometry." International Conference on Machine Learning. PMLR, 2018.
>
> [3] Liang, Tengyuan, and Alexander Rakhlin. "Just interpolate: Kernel “ridgeless” regression can generalize." The Annals of Statistics 48.3 (2020): 1329-1347.
>
> [4] Bietti, Alberto, and Julien Mairal. "On the inductive bias of neural tangent kernels." Advances in Neural Information Processing Systems 32 (2019).

---

> > ### Author Response · Authors · 2022-11-12
> > **Response to Reviewer qnUF (part 2)**
> >
> > ### 4.
> > Regarding the OOD generalization, we would like to add that:
> >
> > * Comparing Figure A.4 to A.5, we can see that the complexity measurements for SimCLR are significantly larger (almost 10 times) than those for $t$-SimCLR, despite sharing very similar trends. This validates our argument in Section 3.1.4, where we stated that uniformity in the unit-sphere will push close clusters farther and pull distant clusters closer, resulting in a less smooth feature map. Such a property can be beneficial to in-distribution classification but harmful when facing OOD data. The phenomenon we observed in Figure 1 in the Gaussian mixture case can also be seen in the CIFAR-10 case.
> > A smaller complexity often indicates a smoother function, and in turn, more robust to input perturbations. Hence, we expect $t$-SimCLR with Euclidean feature space to be better at OOD data.
> >
> > * In the added Figure B.8, we ran new experiments to extend the results shown in Figure 2(b). For the CIFAR-10 trained feature extractor, we evaluate its OOD performance by nearest neighbor classification on CIFAR-100. As can be seen in Figure B.8, the gap significantly widens in the CIFAR-10 -> CIFAR-100 setting, where $t$-SimCLR can outperform by 10% (8% vs 18%). In the OOD setting, the feature extractor from our modified $t$-SimCLR can maintain much-separated clusters.

---

> > > ### Comment · Reviewer_qnUF · 2022-11-16
> > > **Thanks for the clarification**
> > >
> > > Thank you so much for your clarification. Most of my concerns have been addressed. And I decided to raise my score.

---

> > > > ### Author Response · Authors · 2022-11-17
> > > > **Response to Reviewer qnUF**
> > > >
> > > > Thank you very much for your time and effort! We value your comments and will keep working on developing more formal technical statements.

---

### Official Review · Reviewer_RZus · 2022-10-25

**Confidence:** 4
**Correctness:** 3
**Technical Novelty And Significance:** 3
**Empirical Novelty And Significance:** 3
**Recommendation:** 8

**Clarity, Quality, Novelty And Reproducibility:**

The paper is well-written and easy to follow. It is novel to interpret SSCL as SNE, and especially the authors proposed two effective variants of SSCL based on this novel perspective.

**Strength And Weaknesses:**

Strength:

(1) A novel perspective to interpret SSCL as SNE.

(2) The case study based on the Gaussian mixture model for implicit bias and OOD is impressive.

(3) The two variants of SSCL based on SNE are reasonable, which is interesting for exploring OOD generation of SSCL.

Weaknesses:

(1) It is claimed that "Can SNE be revived in the modern era by incorporating SSCL?". It is not discussed in the main manuscripts.

(2) Eq.3.1 is not a good choice. Actually, the conditional probability P_{j|i} is not normalized. It would be a better choice to add the coefficient term 1/2n  as the normalization term for summarization over j.

(3) How are the negative samples constructed in the case study in Sect.3, such as Gaussian mixture setting?

(4) This whole paper claimed to connect SSCL to SNE, but conducted experiments on MoCo-v2, which is not typically an SSCL method as it does not require negative samples. The connection between SNE and MoCo should be discussed as well.

Some minor typos:

(1) "two random sampled data as negative pairs", there are more than two negative pairs.

(2) "is the is a temperature parameter."

(3) In Eq.3.1, it should be $\tilde{x}_i$ and $\tilde{x}_j$


**Summary Of The Paper:**

A novel perspective of self-supervised contrastive learning (SSCL) is investigated. To be specific, this paper interprets SSCL as stochastic neighbor embedding (SNE), and studies implicit bias and out-of-distribution generalization under this framework. The case study about OOD is interesting to me, and also the extension of SSCL based on SNE is impressive.

**Summary Of The Review:**

Overall, this paper is interesting and impressive. If the weaknesses paints can be well addressed, it would be better.

---

> ### Author Response · Authors · 2022-11-12
> **Response to Reviewer RZus**
>
> Thank you very much for the helpful comments. Below we address the points you raised in the review.
>
> ###  Can SNE be revived in the modern era by incorporating SSCL?
> Thank you for the question. In our work, we mainly showcased how we can borrow existing works on SNE to improve SSCL.
> From our perspective, the InfoNCE-type SSCL and SNE are fundamentally the same. In view of (S1) and (S2), SSCL mainly focuses on how to specify input space similarities (by augmentation), while SNE mainly focuses on how to preserve such a pairwise similarity in the feature space.
> In a way, SSCL can be viewed as the modern SNE for complicated image data.
> Thus, we believe that introducing $t$-SNE to SimCLR is also a way of reviving SNE, and the resulting $t$-SimCLR can be viewed as an improved $t$-SNE as well.
> Nevertheless, standard SNE methods on more complicated data can be improved by incorporating data augmentations on top of pre-defined distances.
> We have added a paragraph in the Discussion section.
>
> ### Eq.(3.1)
> Thank you so much for the advice! We realized that the probability is not normalized and we have changed the expression in the draft following your suggestion.
>
> ### How are the negative samples constructed
> The negative samples are the same as standard SimCLR training, i.e., in one batch, for one anchor, its negative pairs are all samples that are not its positive pair. We have added this clarification in the Gaussian mixture setting paragraph.
>
> ### SNE and MoCo
> SNE and MoCo share the same objective function and the main difference is that MoCo utilizes a memory bank to store the features as negative samples. MoCo-v2 is still a typical SSCL method with some improvement for alleviating the need for large batch sizes and therefore eases large-scale training.
>
> ### Some minor typos:
> Thanks for pointing them out! we have corrected them in the updated draft.

---

> > ### Comment · Reviewer_RZus · 2022-11-15
> > **Thanks for the authors' clarification**
> >
> > I am still not convinced by the answers regarding the explanation "Can SNE be revived in the modern era by incorporating SSCL?"
> >
> > Although InfoNCE-type SSCL and SNE can be viewed the same to some extent, their targets are still different. I do not agree with the claim that "the resulting t-SimCLR can be viewed as an improved t-SNE as well." First, the superiority of t-SimCLR is not discussed from the perspective of representation learning; Second, empirical comparisons between these two methods are also required for better understanding.

---

> > > ### Author Response · Authors · 2022-11-17
> > > **Response to Reviewer RZus**
> > >
> > > Thanks for raising this point!
> > >
> > > Indeed, SNE methods are mainly used for data visualization. By choosing the feature space dimension to be 2, various SSCL methods can also be used for data visualization.
> > > We added Appendix B.7 to showcase the 2D visualization of CIFAR-10 for the standard $t$-SNE, SimCLR, and $t$-SimCLR.
> > > CIFAR-10 is challenging for $t$-SNE and it can barely reveal any clusters (Figure B.10) while our t-SimCLR produces much more separation among different labels (Figure B.11, the nearest neighbor classification accuracy on CIFAR-10 test data is 56.6%).
> > > In Figure B.12, we visualize the outcome from SimCLR (the nearest neighbor classification accuracy on CIFAR-10 test data is 24.8%).
> > > We believe there are more investigations to be made in this direction.

---

### Official Review · Reviewer_DMAs · 2022-10-28

**Confidence:** 3
**Correctness:** 3
**Technical Novelty And Significance:** 3
**Empirical Novelty And Significance:** 3
**Recommendation:** 6

**Clarity, Quality, Novelty And Reproducibility:**

The paper is reasonably clear, and the ideas seem novel. The statements are reproducible from the proofs, however, there are few hand-wavy statements without proper proofs.

Suggestions for improvement:
- More formal technical writing, with exact statements. Clarifying when the statement is about a toy setting vs in general
- Empirical evidence to show that $C(f)$  is minimized
- Adding discussion on recent relevant work (mentioned above)
- In Fig B.4, are the labels incorrect? It seems that SimCLR is doing better than $t$-SimCLR. Fix this?

**Strength And Weaknesses:**

**Strengths**:
- The connection to SNE, although intuitive, is new and provides useful insights, which the paper explores through different angles.
- Performance gains from the suggested algorithmic improvements are pretty impressive!

**Weaknesses**:
- Theoretical results in the paper are not clearly presented, and seem more like intuitions than formal statements. For example, in section 3.1.1, the distance proposition is limited to only the simplistic gaussian noise injection setup, and the difference with mixup is not formalized. Same holds for section 3.1.2. Section 3.1.3 claims that “SSCL exhibit neighbor-preserving property and we identify it as an implicit bias”, but the corollary assumes that if $C(f)$ is minimized then show distances are preserved. It is not clear to me why $C(f)$ would be minimized by SSCL.
- Comparison to prior work on relating SSCL to spectral clustering and inductive bias of function classes is not discussed (see HaoChen et al. 2021, HaoChen et al. 2022, Saunshi et al. 2022).

**Summary Of The Paper:**

The paper shows a connection between self-supervised contrastive learning (SSCL) and stochastic neighborhood embedding (SNE), a data visualization method based on preserving distances. More formally, the authors show that SSCL is a form of SNE with pairwise distance/similarity defined by the data augmentation. Leveraging this connection and working with a simple mixture of gaussians setting, the authors provide theoretical insights into
- performance of domain-agnostic augmentations such as mixup
- uniformity and alignment of the learned features
- *expected Lipschitz constant* as an implicit bias of SSCL
- out of distribution robustness with normalized versus unnormalized features

These theoretical results are accompanied with new practical ideas that give significant improvements over standard SSCL. These ideas are:
- using a weighted loss with the weighting dependent on the augmentation pair
- $t$-SSCL in the spirit of $t$-SNE by replacing the gaussian distribution to a heavy tail $t$-distribution

Lastly, the paper show the benefits of these modifications in large scale experiments in terms of domain transfer and o.o.d. generalization.

**Summary Of The Review:**

Despite the issues with the technical content in the paper, the practical advantages of the proposed viewpoints are pretty strong. Therefore, I lean towards acceptance. I would encourage the authors to clean up the theoretical sections to improve the paper.

---

> ### Author Response · Authors · 2022-11-12
> **Response to Reviewer DMAs**
>
> Thank you very much for the helpful comments. Below we address the points you raised in the review.
> ### More formal technical writing
> You are right that the proposition in Section 3.1.1. is restricted to Gaussian noise injection. However, the analysis in Section 3.1.2. and 3.1.3. is general and does not rely on the Gaussian assumption. We have added one paragraph at the beginning of Section 3.1 to make the clarification. Please also see the Overall Response 2 for more.
>
> Specifically, in Section 3.1.3, it is true that we only hypothesized, and did not prove $C(f)$ to be minimized implicitly during training. Theorem 3.5 and Corollary 3.6 are all derived with the assumption that $C(f)$ is minimized. On one hand, following your suggestion, we have conducted numerical experiments to investigate whether $C(f)$ is being minimized during training and the results are supportive.
> On the other hand, we observed that $C(f)$ correlates well with other more common complexity measurements such as $l_2$-norm, which is usually shown to be implicitly minimized in regression tasks [1,2].
> We believe that the $C(f)$ is minimized implicitly and we will seek to prove it for future work.
>
> Reference:
>
> [1] Gunasekar, Suriya, et al. "Implicit bias of gradient descent on linear convolutional networks." Advances in Neural Information Processing Systems 31 (2018).
>
> [2] Gunasekar, Suriya, et al. "Characterizing implicit bias in terms of optimization geometry." International Conference on Machine Learning. PMLR, 2018.
>
>
> ### Empirical evidence on $C(f)$ being minimized
>
> We have added new experiments investigating the complexity $C(f)$ during the training process, in both the Gaussian mixture setting and the CIFAR-10 setting.
> The results are shown in the added Section A.2, with added Figures A.4 and A.5. Please see the Overall Response 1.1 for the details.
>
>
> ### Related works:
> Thank you for pointing out the related works.
> The main difference between our work and spectral contrastive learning is that we focus on different aspects of contrastive learning.
> More specifically:
> * In [HaoChen et al. 2021], the contrastive learning loss has to be modified to a very specific form, in order to take advantage of the graph literature.
> Their argument cannot be directly applied to the InfoNCE loss. In comparison, our analysis directly applied to InfoNCE, which is the most popular contrastive learning loss.
>
> * The works by HaoChen et al. made assumptions on data augmentation and data distribution. Our work focuses on understanding the feature learning process of SSCL through the SNE perspective, studying how the input space similarity is preserved in the feature space. We do not study the data augmentation property or the generalization ability on classification.
>
> We have included [HaoChen et al. 2022] in the related work.
>
>
> [Saunshi et al. 2022] discussed several sources of inductive bias in contrastive learning and they specifically focused on the function class and algorithm with rigorous analysis under the linearity assumption.
> In our work, we aim to understand SSCL from the SNE perspective and the implicit bias characterized by SNE with uniformity is only part of the insights, which can be seen as induced by the training algorithm and is not discussed in previous works.
> We have added a discussion about [Saunshi et al. 2022] in Section 3.1.3.
>
> ### Figure B.4
> Thanks for pointing it out. We made a mistake when labeling and $t$-SimCLR should be on the top. We have changed the figure.

---

> > ### Comment · Reviewer_DMAs · 2022-11-15
> > **Thanks for the response!**
> >
> > Thank you for responding to my questions/concerns.
> > - I appreciate the added experiments for measuring $C(f)$, they help address some of my doubts regarding the assumption in the theoretical parts.
> > - I do not think the authors have sufficiently addressed my concerns regarding technical writing. Exact statements are still missing and as I mentioned, the sections seem more intuition based than formal. I encourage the authors to spend some cycles on improving the presentation.
> > - Regarding related work, thanks for addressing these.
> >
> > I will maintain my score.

---

> > > ### Author Response · Authors · 2022-11-17
> > > **Response to Reviewer DMAs**
> > >
> > > Thank you very much for your time and effort! We value your advice and will keep working on developing more formal technical statements.

---

### Official Review · Reviewer_emn1 · 2022-10-28

**Confidence:** 4
**Correctness:** 4
**Technical Novelty And Significance:** 4
**Empirical Novelty And Significance:** 4
**Recommendation:** 8

**Clarity, Quality, Novelty And Reproducibility:**

- Clarity: The writing is easy to follow. It’s a little dense but I guess it’s ok.
- Quality: The quality is good. The provided insights are significant. The analysis is well supported by numerical experiments.
- Novelty: The novelty is high. The connection between SNE and InfoNCE type SSCL is new to me and the analysis that followed is original and natural.

**Strength And Weaknesses:**

Strength:
The novelty is strong. The connection between SimCLR and SNE is rarely discussed and the SNE perspective of SSCL is novel to me. The connection is presented in an inspiring way where the authors provided several new insights and demonstrated several SNE inspired modifications to SimCLR. While mainstream self-supervised learning focuses on methodology and benchmark performance, it’s nice to see that this work started by investigating the Gaussian mixture setting. Though oversimplified, the low-dimensional simulated setting does provide plenty of insights and can be explicitly analyzed. The acquired insights are further validated by the following real-data experiments.

The implicit bias towards SNE with uniformity constraint (sec 3.1.3) is most interesting to me, which points to a very important, but perhaps overlooked question in understanding contrastive learning. Contrastive learning works extremely well in practice, which is surprising since the learning objective is so simple and has (infinitely many) trivial solutions. I agree with the authors that the discriminative features are mainly learned implicitly. The order preserving phenomenon does shed light on understanding the CL for real data.
Following the authors’ argument, the role of positive pair alignment is merely to simplify the input space, by merging equivalent classes. Then the feature learning process is essentially SNE with uniformity constraint.  This is a very interesting point to me and may call for more investigations.

The dimensional efficiency (sec 4.2) and OOD generalization are also interesting and are very thought-provoking. In Figure2(b), it’s surprising to see that the dimensionality gains for CIFAR-10 classification are almost non-existent after only 16. It is interesting to see how the dimension affects the performance in the OOD setting.

Weaknesses:
The title may have been overclaimed. The paper only concerns the InfoNCE loss (SimCLR, MoCo) and there are various other self-supervised learning methods, e.g., SimSiam, BYOL, etc. How does the connection to SNE carry over to them? Can we observe the same phenomenon?

Some questions/comments:
- I am curious to see how does feature dimensions affect the performance in the OOD setting. My guess is that the performance gain due to higher dimensions should be much more significant than IID case, e.g., Figure 2(b).

- The order/neighbor preserving phenomenon is not very easy to see in the 2-d Gaussian case. I would suggest the authors make the clusters more extreme so that which should be closer to which is easier to see.

- Typo: The first sentence of sec 3.1.3, results on SSCL **provide**


**Summary Of The Paper:**

This paper studies contrastive self-supervised learning from the view of SNE. The authors show that InfoNCE loss can be seen as a special case of SNE, by setting the data similarity matrix P according to the data augmentation and embedding matrix Q as the softmax of pairwise similarity. Under the SNE framework, new insights for domain-agnostic data augmentation, implicit bias and OOD generalization are presented. Moreover, from SNE perspective, this paper proposes to improve SimCLR by 1) introducing weighted matrix P to better utilize the augmentation detail and 2) t-SNE style InfoNCE loss, i.e, l2 instead of cosine to improve OOD generalization and exponential to polynomial to improve feature dimension efficiency. Various experiments show that improvement can be achieved in both in-distribution and out-of-distribution generalization.


**Summary Of The Review:**

This paper is novel and original. The contribution is solid. The SNE perspective is helpful for both theoretical understanding of SSCL as well as guidance for practical improvement. I believe this work is helpful for both the SNE and SSCL communities and may facilitate further exchange between the two.

---

> ### Author Response · Authors · 2022-11-12
> **Response to Reviewer emn1**
>
> Thank you very much for the helpful comments.
>
> We are glad that you liked our analysis of SNE as an implicit bias of SSCL. Indeed, we believe that one of the main roles that positive pairs play is to simplify the input space, by specifying similarity or invariance.
> If the positive pairs can be perfectly aligned, the feature learning process is mainly determined by the implicit bias within uniformity.
> One particular case where this perspective can help is multi-modal contrastive learning like CLIP [Radford et al. 2021].
> The training process of CLIP is merely aligning the feature space of both modalities, i.e., text feature and image feature of the same underlying object, and offers no direct control on the feature mapping marginally of each of the domains.
> Our SNE perspective can potentially help us understand why the image features obtained from CLIP can give a good discriminative performance.
>
>
> ### "The title may have overclaimed"
>
> We agree that our analysis only applies to the SSCL methods that utilize the InfoNCE loss.
> As acknowledged in the Discussion section, "Our analysis has limitations and the insights from SNE are not universally applicable for all SSCL methods, e.g., Zbontar et al. (2021); Yang et al. (2021) don’t fit in our framework."
> However, as InfoNCE is the most common loss for contrastive learning, we believe that our insights can be widely applied.
>
>
> ### "I am curious to see how does feature dimensions affect performance in the OOD setting."
>
> Thank you for this interesting question! We conducted new experiments to evaluate the OOD performance, based on the existing CIFAR-10 results in Figure 2(b). Please see the Overall Response 1.2 for more details.

---

### Author Response · Authors · 2022-11-12
**Overall Response and Updates in Revision**

We thank all the reviewers for their helpful comments! We have updated the draft, both the main file and the appendix. All modifications are highlighted in the color blue. Below we address some common concerns.


## 1. Added Experiments
### 1.1. Empirical evidence on $C(f)$.
$C(f)$ is the expected Lipschitz constant of $f$ defined in Equation (3.2) of the paper, where we showed in Corollary 3.6 that minimizing $C(f)$ leads to preserving pairwise distance. We hypothesized $C(f)$ to be minimized implicitly during training.
We further provide empirical evidence by adding new experiments investigating the complexity $C(f)$ during the training process, in both the Gaussian mixture setting and the CIFAR-10 setting.
The results are shown in the added Section A.2 in the updated draft, with added Figures A.4 and A.5.


In the Gaussian mixture setting, the feature extractor is a fully connected ReLU network. Besides $C(f)$, we also evaluated the popular sum of squared weights. The observations on SimCLR are listed below:
* The expected Lipschitz constant $C(f)$ is small in initialization. It first increases (till around 100 iterations) and then consistently decreases. This empirically supports the implicit bias towards minimizing $C(f)$.
* $C(f)$ and the sum of squared weights share very similar patterns.
* The SNE loss is non-increasing, as if we are doing stochastic neighbor embedding using $l_2$-distance. This further supports the implicit bias as SNE with uniformity constraint.


In the CIFAR-10 case, the feature extractor is ResNet-18 plus a fully-connected projection layer. The output from ResNet-18 is usually called representation (512 dimensional) and is utilized for downstream tasks while the projection (128-dimension) is used for training. Note that the feature extractor discussed in this work is the projection layer that the training objective directly optimizes. Besides $C(f)$, we also evaluate the $l_2$-norm of the representation.
The observations for SimCLR and $t$-SimCLR on CIFAR-10 are summarized below:
* $C(f)$ for the projection layer shares similar patterns as in the Gaussian mixture case, first increasing and then decreasing. However, $C(f)$ for the representation layer monotonically decreases.
* $C(f)$ for the projection layer and the $l_2$-norm in the representation layer share almost identical patterns.
* Comparing SimCLR, both the the calculated $C(f)$ and $l_2$-norm are much smaller for $t$-SimCLR.


In conclusion, on one hand, our empirical results demonstrate that the complexity of the feature extractor $C(f)$ does decrease during training and seems to be implicitly minimized. On the other hand, its trend is shared with other more popularly used complexity measurements. Finally, all complexity measurements for SimCLR appear significantly larger than those for $t$-SimCLR.

### 1.2. Dimension Efficiency in OOD setting
We conducted new experiments to evaluate the OOD performance, based on the existing CIFAR-10 results in Figure 2(b). For each of the models (SimCLR and $t$-SimCLR with different feature dimensions), we further evaluate the nearest neighbor classification accuracy for CIFAR-100 test data. We added one paragraph in Section B.3 with the CIFAR-10 to CIFAR-100 results plotted in the added Figure B.8 in the updated appendix.
What we found is that
* The gain of extra dimensions in the OOD case does vanish later than that in the in-distribution case.
* The advantage of SimCLR vs. $t$-SimCLR is significant with around 10\% improvement on nearest neighbor accuracy with $d=128$. This indicates that our proposed $t$-SimCLR produces better-separated clusters.

### 1.3. Data Visualization in 2D
By choosing the feature space dimension to be 2, various SSCL methods can also be used for data visualization. We added Appendix B.7 to showcase the 2D visualization of CIFAR-10 for the standard $t$-SNE, SimCLR, and $t$-SimCLR.

## 2. Clarification of Analysis
In this work, we often start by showing insights in the simple Gaussian mixture setting. We then formulate our analysis and demonstrate that the same insights can be carried over to real applications, e.g., image classification, by empirical evaluation.
Though we often use the simple Gaussian mixture for illustration, it should be noted that all our formal results (Theorem 3.5, Corollary 3.6, etc.) are general and do not depend on the Gaussian assumption. Due to the page limit, we also put some formal theoretical results in the appendix, e.g., Theorem A.3 to analyze the alignment and uniformity of $t$-SimCLR.

The main goal of this work is to provide new insights of SSCL from the perspective of SNE.
We agree that our work can definitely benefit from more rigorous analysis.
For instance, bridging the gap between the simple Gaussian mixture setting and real image data would certainly be useful, but it incurs enormous technical difficulty.
We sincerely hope our SNE perspective can inspire more investigations along this direction.

---

### Public Comment · ~Dmitry_Kobak1 · 2023-02-03
**Two related ICLR-2023 papers**

Congratulations on acceptance! I'm happy to mention that there are two related ICLR-2023 accepted papers:

1) "From t-SNE to UMAP with contrastive learning" (https://openreview.net/forum?id=B8a1FcY0vi) gives a contrastive learning perspective on t-SNE, and in particular uses the InfoNCE loss function (with mini-batches) to optimize t-SNE. We call it InfoNC-t-SNE (we have a non-parametric and also a parametric implementation; the parametric version optimizes a neural network).

2) "Unsupervised visualization of image datasets using contrastive learning" (https://openreview.net/forum?id=nI2HmVA0hvt) suggests something similar to your t-SimCLR with 2D output, and develops an effective training approach that is able to generate high-quality 2D visualizations (we call it t-SimCNE), e.g. of CIFAR-10 and CIFAR-100. This is strongly related to what you do in the Appendix B.7 and show in Figure B.11.

---

### Decision · Program_Chairs · 2023-01-20

**Decision:**

Accept: poster

**Justification For Why Not Higher Score:**

The theoretical arguments are often hand-wavy.

**Justification For Why Not Lower Score:**

The established connection between contrastive learning and SNE is valuable.

**Metareview: Summary, Strengths And Weaknesses:**

This paper studies contrastive self-supervised learning from the view of SNE.

Strengths:
- the connection between contrastive learning and SNE is valuable
- the paper is well written

Weaknesses:
- the theoretical arguments are often hand-wavy (pointed out by 2 reviewers)

I recommend acceptance as a poster. I would encourage the authors to clean up the theoretical sections to improve the paper.

**Note From Pc:**

if the above contains the word "oral" or "spotlight" please see: "oral" presentation means -> notable-top-5% and "spotlight" means -> notable-top-25%. As stated in our emails, we are disassociating presentation type from AC recommendations